# Bayesian and non-bayesian analysis for stress-strength model based on progressively first failure censoring with applications

Salem A. Alyami[1☯], Amal S. Hassan[2☯], Ibrahim Elbatal[1], Olayan Albalawi[3], Mohammed Elgarhy[4,5]*, Ahmed R. El-Saeed[1,6]

1 Department of Mathematics and Statistics, Faculty of Science, Imam Mohammad Ibn Saud Islamic University (IMSIU), Riyadh, Saudi Arabia, 2 Faculty of Graduate Studies for Statistical Research, Cairo University, Giza, Egypt, 3 Department of Statistics, Faculty of Science, University of Tabuk, Tabuk, Saudi Arabia, 4 Mathematics and Computer Science Department, Faculty of Science, Beni-Suef University, Beni-Suef, Egypt, 5 Department of Basic Sciences, Higher Institute of Administrative Sciences, Belbeis, AlSharkia, Egypt, 6 Department of Basic Sciences, Obour High Institute for Management & Informatics, Al Qalyubia, Egypt

☯ These authors contributed equally to this work.
* m_elgarhy85@sva.edu.eg

**Data Availability Statement:** All relevant data are within the manuscript and its Supporting Information files.

## Abstract

This article examines the estimate of $\vartheta = P[T < Q]$, using both Bayesian and non-Bayesian methods, utilizing progressively first-failure censored data. Assume that the stress ($T$) and strength ($Q$) are independent random variables that follow the Burr III distribution and the Burr XII distribution, respectively, with a common first-shape parameter. The Bayes estimator and maximum likelihood estimator of $\vartheta$ are obtained. The maximum likelihood (ML) estimator is obtained for non-Bayesian estimation, and the accompanying confidence interval is constructed using the delta approach and the asymptotic normality of ML estimators. Through the use of non-informative and gamma informative priors, the Bayes estimator of $\vartheta$ under squared error and linear exponential loss functions is produced. It is suggested that Markov chain Monte Carlo techniques be used for Bayesian estimation in order to achieve Bayes estimators and the associated credible intervals. To evaluate the effectiveness of the several estimators created, a Monte Carlo numerical analysis is also carried out. In the end, for illustrative reasons, an algorithmic application to actual data is investigated.

## 1 Introduction

In lifetime experiments and reliability studies, it is common for data to not be entirely captured owing to constraints in time, money, or resources, or because of staffing changes and accidents. Often, censored samples are employed in these situations. These days, a variety of censorship techniques are used for lifetime assessments. Type-I censoring (TI-C) and Type-II censoring (TII-C) are two of the most commonly used methods. The test ends in TI-C when it reaches the predetermined time. When a certain number of units have failed in TII-C, the test is finished. There is no flexibility about the removal of items at stages other

**Funding:** This work was supported and funded by the Deanship of Scientific Research at Imam Mohammad Ibn Saud Islamic University (IMSIU) (grant number IMSIU-RPP2023003).

**Competing interests:** The authors have declared that no competing interests exist.

than the test's final step under TI-C, and TII-C systems, which is a typical drawback. The progressive TII-C (POTII-C) system was first presented in the literature to address this issue. Consult Balakrishnan and Aggarwala [1] for additional information. Despite the fact that the experimental efficiency under POTII-C can be greatly increased, the test's length is still too long. Thus, Johnson [2] presented a first-failure censoring technique in which the experimenter could be interested in grouping test units into many groups, each group consisting of collection test units, and running each group's test units concurrently until each group experiences its first failure. Groups cannot be excluded from the test before the final termination stage due to first-failure censorship. But in real life, this is much more desired. This censorship strategy works best in situations where a product has a long lifespan, there are few inspection facilities, and the examination material is reasonably priced [3]. For further information, check, for instance, [4, 5]. Indeed, the groups that were presented throughout the test cannot be eliminated due to the first-failure censorship. Thus, a novel life test known as progressive censoring with first-failure censoring scheme (POFIF-CS), which combines progressive censoring with first-failure censoring, was introduced by Wu and Kus [6] in order to get around this problem and enhance test efficiency. It is an advancement and expansion of progressive censorship, and because of its adaptability, it is frequently employed in experimental design.

The POFIFCS is explained as follows: A life test is applied to $n$ independent groups with $h$ elements per group. The $\mathscr{R}_1$ groups and the group with the first failure observation are randomly eliminated from the test when the first failure happens $Q_{1:m:n:h}$. Similarly, $\mathscr{R}_2$ groups and the group that includes the second failure, $Q_{2:m:n:h}$, are withdrawn as soon as it appears, and so on. Once the $m$th failure occurs, the other $\mathscr{R}_m$ groups are also eliminated, along with the group that includes the $m$th failure observation. With the progressive censoring scheme $\mathscr{R} = (\mathscr{R}_1, \mathscr{R}_2, \ldots, \mathscr{R}_m)$, where $n = m + \mathscr{R}_1 + \mathscr{R}_2 + \ldots + \mathscr{R}_m$, the observed failure times, $Q_{1:m:n:h} < Q_{2:m:n:h} < \ldots < Q_{m:m:n:h}$ are referred to as the POFIF-CS. It may be shown that the POFIF-CS is reduced to a first-failure censoring scheme in a particular scenario when $\mathscr{R}_1 = \mathscr{R}_2 = \ldots = \mathscr{R}_m = 0$. Similarly, first-failure TII-C scheme is a specific instance of this censoring method where $\mathscr{R}_1 = \mathscr{R}_2 = \ldots = \mathscr{R}_{m-1} = 0$ and $\mathscr{R}_m = n - m$. Suppose that $q_{1:m:n:h}, q_{2:m:n:h}, \ldots, q_{m:m:n:h}$ be the observed failure times under investigation from a continuous function, then the joint probability density function (PDF) is provided via:

$$g_{Q_{1:m:n:h}, \ldots, Q_{m:m:n:h}}(q_1, \ldots, q_m) = Dh^m \prod_{i=1}^{m} g(q_{i:m:n:h})[\bar{G}(q_{i:m:n:h})]^{h(\mathscr{R}_i+1)-1}, \tag{1}$$

where $D = \prod_{i=0}^{m}(n - i - \sum_{h=0}^{i} \mathscr{R}_h)$, $g(.)$ is the PDF, and $\bar{G}(.)$ is the survival function (SF).

One of the most fascinating areas of reliability theory is stress-strength (SS) models. The SS reliability (SSR) parameter, or $\vartheta = P[T < Q]$ is commonly employed to assess performance when a system with strength $Q$ is exposed to stress $T$. When the employed strength in an operating system surpasses its stress, the system becomes reliable; if not, it malfunctions. In statistical research, estimation of $\vartheta$ has been a topic of interest since Birnbaum's 1956 study. Since then, the aforementioned model has been frequently utilised in industrial engineering, economics, psychology, and medical research. In the statistical literature, the issue of SS model estimation has gotten a lot of attention. A military application of $\vartheta$ and an example application of $\vartheta$ in rocket engines were discussed by Refs. [7, 8]. Kotz et al. [9] provided a comprehensive book on the various SSR models.

Numerous scholars have conducted recent work on the SSR model for independent random variables under various scenarios, including complete samples, record values, and ranked set sampling. These scholars include Jia et al. [10], Babayi and Khorram [11], Kizilaslan and Nadar [12], Al-Omari et al. [13], Almarashi et al. [14], Hassan et al. [15, 16], Alsadat et al. [17] among others. Lately, there has been a lot of focus on researching the estimation of $\vartheta$ under censored data. For instance, the SSR estimation for the Burr XII (BXII) distribution was examined by Lio and Tsai [18] using POFIF-CS. The Lindley distribution's SS parameter estimation using POFIF-CS was covered by Kumar et al. [19]. For independent generalized inverted exponential populations, Krishna et al. [20] examined the estimation of SSR using POFIF-CS. Byrnes et al. [21] handled the Bayesian inference of SSR when the stress and strength random variables have a BXII distribution. Krishna et al. [22] examined estimation of SSR based on POFIF-CS data from two independent inverse Weibull distributions. The SSR estimates under POFIF-CS for independent generalized Maxwell populations were covered by Saini et al. [23].

Burr [24] proposed twelve different forms of cumulative distribution functions (CDFs). Type XII and Type III have received particular attention when modelling lifetime or survival data. It is important to remember that the Burr III (BIII) distribution is reciprocal of Burr XII (BXII) distribution. The BIII distribution is more adaptable and contains a range of different skewness and kurtosis levels. Numerous statistical modelling fields, including forestry [25], meteorology [26], and reliability [27], modeling crop rice [28], fracture roughness data [29] have found extensive use for this distribution. The two parameter BXII distribution is widely utilised in the domains of failure time and life time modelling. The PDF and SF of the BXII distribution with shape parameters $v > 0$, and $\delta_1 > 0$, are given by:

$$g(q) = v\delta_1 q^{v-1}(1 + q^v)^{-\delta_1 - 1}, \quad q \in \mathbb{R}^+, \tag{2}$$

and

$$\bar{G}(q) = (1 + q^v)^{-\delta_1}, \quad q \in \mathbb{R}^+. \tag{3}$$

Moreover, the PDF and SF of the BIII distribution are provided below

$$f(t) = v\delta_2 t^{-(v+1)}(1 + t^{-v})^{-\delta_2 - 1}, \quad t \in \mathbb{R}^+, \tag{4}$$

and

$$\bar{F}(t) = 1 - (1 + t^{-v})^{-\delta_2}, \quad t \in \mathbb{R}^+, \tag{5}$$

where $v > 0$, and $\delta_2 > 0$, are the shape parameters. Let $Q$ and $T$ stand for the two independent random variables related to strength and stress that are observed in BXII $(v, \delta_1)$ and BIII $(v, \delta_2)$ distributions, respectively. The SSR parameter is evaluated assuming that the models have different second-shape parameters but the identical first-shape parameter, that is, $Q \sim$ BXII $(v, \delta_1)$ and $T \sim$ BIII $(v, \delta_2)$. Consequently $\vartheta$ is obtained as follows:

$$\vartheta = \int_0^\infty g(q) G_T(q) dq = \int_0^\infty \vartheta \delta_1 q^{\vartheta - 1}(1 + q^\vartheta)^{-\delta_1 - 1}(1 + q^{-v})^{-\delta_2} dq = \left[\frac{\Gamma(\delta_1 + 1)\Gamma(\delta_2 + 1)}{\Gamma(\delta_1 + \delta_2 + 1)}\right], \tag{6}$$

where $\Gamma(.)$ is the gamma function. The SSR $\vartheta$ depends on the shape parameters $\delta_1$ and $\delta_2$.

The main contribution of this work can be described as follows:

- Provide the classical and Bayesian estimates of the reliability parameter $\vartheta$ assuming that $Q$ and $T$ are independent random variables having BXII and BIII distributions, respectively, with common first-shape parameter based on POFIF-CS.

- For the Bayesian estimation approach, two loss functions along with informative (INP) and non-informative (N-INP) priors are taken into consideration.

- For classical method, the asymptotic confidence intervals (Asy-CIs), and normal approximate confidence intervals (NA-CIs) via delta method are obtained. For Bayesian method, the highest probability density (HPD) intervals are determined

- The Markov chain Monte Carlo (MCMC) approach is used to approximate the reliability estimator due to the computing challenges associated with provided posteriors.

- To investigate the behaviour of different estimates, a simulation research is carried out with the accuracy criteria stated under different POFIF-CS and sample sizes.

- Two real-world scenarios are provided to show how the suggested estimating techniques work.

The remainder of the paper is arranged as follows. In Section 2, the maximum likelihood estimate (MLE) of $\vartheta$ is derived based on POFIF-CS. Section 3 presents the Bayesian estimator of $\vartheta$ based on the linear exponential (LINx), and squared error (SE) loss functions. The numerical experiment, which is based on Monte Carlo simulations, is presented in Section 4. Section 5 provides an application example using real data. Section 6 provides the conclusion.

## 2 Maximum likelihood estimator

In this section, we acquire the MLE of $\vartheta = P[T<Q]$, assuming that $Q \sim$ BXII $(v, \delta_1)$ and $T \sim$ BIII $(v, \delta_2)$ where $Q$ and $T$ are independent based on POFIF-CS.

Let $(q_{1:m_1:n_1:h_1}, q_{2:m_1:n_1:h_1}, \ldots, q_{m_1:m_1:n_1:h_1}) = (q_1, q_2, \ldots, q_{m_1})$ be the POFIF-CS from BXII $(v, \delta_1)$ with progressive censoring scheme $\mathscr{R}' = (\mathscr{R}'_1, \ldots, \mathscr{R}'_{m_1})$. Also, let $(t_{1:m_2:n_2:h_2}, t_{2:m_2:n_2:h_2}, \ldots, t_{m_2:m_2:n_2:h_2}) = (t_1, t_2, \ldots, t_{m_2})$ be independent POFIF-CS from BIII $(v, \delta_2)$ with censoring scheme $\mathscr{R}^* = (\mathscr{R}^*_1, \ldots, \mathscr{R}^*_{m2})$. Substituting (2), (3), (4), (5) in (1), the likelihood function (LF) is rewritten as:

$$
\begin{aligned}
L(v, \delta_1, \delta_2) \quad &= D_1 D_2 h_1^{m_1} h_2^{m_2} \\
&\times \prod_{i_1=1}^{m_1} (v\delta_1 q_{i_1}^{v-1}(1+q_{i_1}^v)^{-\delta_1-1})[(1+q_{i_1}^v)^{-\delta_1}]^{h_1(\mathscr{R}'_{i_1}+1)-1} \\
&\times \prod_{i_2=1}^{m_2} (v\delta_2 t_{i_2}^{-v-1}(1+t_{i_2}^{-v})^{-\delta_2-1})[1-(1+t_{i_2}^{-v})^{-\delta_2}]^{h_2(\mathscr{R}^*_{i_2}+1)-1},
\end{aligned}
\tag{7}
$$

where $D_1 = \prod_{i_1=0}^{m_1} (n_1 - i_1 - \sum_{h_1=0}^{i_1} \mathscr{R}'_{h_1})$, and $D_2 = \prod_{i_2=0}^{m_2} (n_2 - i_2 - \sum_{h_2=0}^{i_2} \mathscr{R}^*_{h_2})$. Hence the log LF of (7)

is given by

$$
\begin{aligned}
L^{\bullet\bullet} \propto\; & m_1 \ln(v\delta_1) + m_2 \ln(v\delta_2) + (v-1)\sum_{i_1=1}^{m_1} \ln(q_{i_1}) \\
& - (v+1)\sum_{i_2=1}^{m_2} \ln(t_{i_2}) - (\delta_1+1)\sum_{i_1=1}^{m_1} \ln(1+q_{i_1}^{v}) \\
& - (\delta_2+1)\sum_{i_2=1}^{m_2} \ln(1+t_{i_2}^{-v}) \\
& - \sum_{i_1=1}^{m_1}[(h_1(\mathscr{R}'_{i_1}+1)-1)\delta_1 \ln(1+q_{i_1}^{v})] \\
& + \sum_{i_2=1}^{m_2}[(h_2(\mathscr{R}^*_{i_2}+1)-1)\ln(1-(1+t_{i_2}^{-v})^{-\delta_2})].
\end{aligned}
\tag{8}
$$

To obtain MLEs $\hat{v}, \hat{\delta}_1$, and $\hat{\delta}_2$, we partially differentiating the log-likelihood (8) with respect to $v, \delta_1$, and $\delta_2$, respectively, and then equalizing them to zero, yield

$$
\begin{aligned}
\frac{\partial L^{\bullet\bullet}}{\partial v} = & \frac{m_1+m_2}{\hat{v}} + \sum_{i_1=1}^{m_1} \ln(q_{i_1}) - \sum_{i_2=1}^{m_2} \ln(t_{i_2}) - (\hat{\delta}_1+1)\sum_{i_1=1}^{m_1} \frac{\ln q_{i_1}}{(1+q_{i_1}^{-\hat{v}})} - \sum_{i_1=1}^{m_1} \frac{[h_1(\mathscr{R}'_{i_1}+1)-1]\hat{\delta}_1 \ln q_{i_1}}{(1+q_{i_1}^{-\hat{v}})} \\
& + (\hat{\delta}_2+1)\sum_{i_2=1}^{m_2} \frac{\ln t_{i_2}}{(1+t_{i_2}^{\hat{v}})} - \sum_{i_2=1}^{m_2} \frac{\hat{\delta}_2[h_2(\mathscr{R}^*_{i_2}+1)-1](1+t_{i_2}^{-\hat{v}})^{-\hat{\delta}_2-1}t_{i_2}^{-\hat{v}}\ln t_{i_2}}{[1-(1+t_{i_2}^{-\hat{v}})^{-\hat{\delta}_2}]} = 0,
\end{aligned}
\tag{9}
$$

$$
\frac{\partial L^{\bullet\bullet}}{\partial \delta_1} = \frac{m_1}{\hat{\delta}_1} - \sum_{i_1=1}^{m_1} \ln(1+q_{i_1}^{\hat{v}}) - \sum_{i_1=1}^{m_1}[h_1(\mathscr{R}'_{i_1}+1)-1]\ln(1+q_{i_1}^{\hat{v}}) = 0,
\tag{10}
$$

and

$$
\frac{\partial L^{\bullet\bullet}}{\partial \delta_2} = \frac{m_2}{\hat{\delta}_2} - \sum_{i_2=1}^{m_2} \ln(1+t_{i_2}^{-\hat{v}}) + \sum_{i_2=1}^{m_2} \frac{[h_2(\mathscr{R}^*_{i_2}+1)-1]\ln(1+t_{i_2}^{-\hat{v}})}{[(1+t_{i_2}^{-\hat{v}})^{\hat{\delta}_2}-1]} = 0.
\tag{11}
$$

The MLEs $\hat{v}, \hat{\delta}_1$, and $\hat{\delta}_2$, can be generated from the Eqs (9)–(11) by applying an appropriate iterative process, like the Newton-Raphson technique, for the given values of $(h_1, m_1, n_1, \mathscr{R}'_1, \underline{q})$ and $(h_2, m_2, n_2, \mathscr{R}^*_2, \underline{t})$. Thus, the invariance property of MLEs can be used to generate the MLE $\hat{\vartheta}$ of $\vartheta$ as below:

$$
\hat{\vartheta} = \left[\frac{\Gamma(\hat{\delta}_1+1)\Gamma(\hat{\delta}_2+1)}{\Gamma(\hat{\delta}_1+\hat{\delta}_2+1)}\right].
\tag{12}
$$

Furthermore, the observed Fisher information matrix (FM) is defined as follows for assessing the estimated variance-covariance matrix (VC-M) and associated Asy- CIs of MLEs with

POFIF-CS

$$\hat{V} = \hat{I}^{-1}(\Psi) = - \begin{bmatrix} \dfrac{\partial^2 L^{\bullet\bullet}}{\partial v^2} & \dfrac{\partial^2 L^{\bullet\bullet}}{\partial v \delta_1} & \dfrac{\partial^2 L^{\bullet\bullet}}{\partial v \partial \delta_2} \\[2mm] \dfrac{\partial^2 L^{\bullet\bullet}}{\partial \delta_1 \partial v} & \dfrac{\partial^2 L^{\bullet\bullet}}{\partial \delta_1^2} & \dfrac{\partial^2 L^{\bullet\bullet}}{\partial \delta_1 \partial \delta_2} \\[2mm] \dfrac{\partial^2 L^{\bullet\bullet}}{\partial \delta_2 \partial v} & \dfrac{\partial^2 L^{\bullet\bullet}}{\partial \delta_2 \partial \delta_1} & \dfrac{\partial^2 L^{\bullet\bullet}}{\partial \delta_2^2} \end{bmatrix} \begin{matrix} v = \hat{v} \\ \delta_1 = \hat{\delta}_1 \\ \delta_2 = \hat{\delta}_2 \end{matrix} = \begin{bmatrix} I_{11} & I_{12} & I_{13} \\ I_{21} & I_{22} & I_{23} \\ I_{31} & I_{32} & I_{33} \end{bmatrix} \begin{matrix} v = \hat{v} \\ \delta_1 = \hat{\delta}_1 \\ \delta_2 = \hat{\delta}_2 \end{matrix} = \begin{bmatrix} \hat{I}_{11} & \hat{I}_{12} & \hat{I}_{13} \\ \hat{I}_{21} & \hat{I}_{22} & \hat{I}_{23} \\ \hat{I}_{31} & \hat{I}_{32} & \hat{I}_{33} \end{bmatrix}.$$

Take note that the Appendix (1) contains the equations for second-order partial derivatives. The $(1 - \rho)100\%$ Ay-CIs for $\Psi = (v, \delta_1, \delta_2)$ obtained by utilising the approximated standard normal distribution are provided by

$$\hat{\Psi} \pm Z^{\bullet}_{\rho/2} \sqrt{\widehat{\mathrm{var}}(\hat{\Psi})},$$

where $Z^{\bullet}_{\rho/2}$ denoted the upper $\rho/2$ percent point of standard normal distribution. We also need to find their variances in order to get the Asy-CIs for SSR. We employ the delta approach described in [30] to get an approximation of $\vartheta$. This approach allows the variance of $\vartheta$ and to be roughly calculated as

$$\mathrm{var}(\hat{\vartheta})) = [\Im]^T [\hat{V}][\Im],$$

where, $\Im = \left( \frac{\partial \vartheta}{\partial v}, \frac{\partial \vartheta}{\partial \delta_1}, \frac{\partial \vartheta}{\partial \delta_2} \right)$. Thus, the two-sided $100(1 - \rho)\%$ Asy-CI of $\hat{\vartheta}$, can be constructed as follows:

$$\hat{\vartheta} \pm Z^{\bullet}_{\rho/2} \sqrt{\widehat{\mathrm{var}}(\hat{\vartheta})}.$$

where

$$\frac{\partial \vartheta}{\partial v} = 0,$$

$$\frac{\partial \vartheta}{\partial \delta_1} = \frac{\Gamma(\delta_2 + 1)[\Gamma'(\delta_1 + 1)(\Gamma(\delta_1 + \delta_2 + 1)) - \Gamma(\delta_1 + 1)(\Gamma'(\delta_1 + \delta_2 + 1))]}{(\Gamma(\delta_1 + \delta_2 + 1))^2},$$

and

$$\frac{\partial \vartheta}{\partial \delta_2} = \frac{\Gamma(\delta_1 + 1)[\Gamma'(\delta_2 + 1)(\Gamma(\delta_1 + \delta_2 + 1))\Gamma(\delta_2 + 1)(\Gamma'(\delta_1 + \delta_2 + 1))]}{(\Gamma(\delta_1 + \delta_2 + 1))^2},$$

where, $\Gamma'(.)$ is the first derivative of gamma function. Further, $\frac{\hat{\vartheta} - \vartheta}{\sqrt{\widehat{\mathrm{var}}(\hat{\vartheta})}}$ follow standard normal distribution asymptotically.

## 2.1 Normal approximation of the log-transformed of $\vartheta$

One limitation of the $100(1 - \rho)\%$ Asy-CIs, as discussed earlier, is that they can yield negative lower bounds for parameters that are inherently positive. In such cases, substituting a negative value with zero might be considered. To address this shortcoming in the accuracy of the normal approximation, Meeker and Escobar [31] proposed a solution in the form of log-transformed MLE-based Asy-CIs. Their findings suggest that this type of confidence interval offers

improved coverage probability. The NA-CI, which is a $100(1 - \rho)\%$ NA-CI, is calculated for log-transformed MLEs and can be expressed as follows:

$$\hat{\vartheta} \times e^{\pm \frac{z_{\rho/2} \sqrt{\widehat{var(\hat{\vartheta})}}}{\hat{\vartheta}}}$$

## 3 Bayesian estimation

Bayesian inference has been increasingly popular in various industries such as engineering, clinical medicine, biology, and others. The ability to analyze data using prior information makes it highly beneficial in reliability studies, particularly when data availability is a significant concern. The Bayes estimate (BE) and accompanying credible intervals of $\vartheta$ can be calculated using the POFIF-CS method in this section. The BE of $\vartheta$ is considered using the SE and LINx loss functions, with the INP and N-INP. Utilizing alternative gamma priors is straightforward and can lead to more expressive posterior density estimates due to the ability of the gamma distribution to assume varied shapes based on the parameter values. Consequently, we conducted an examination of gamma density priors, which possess greater flexibility compared to other complex prior distributions. Given that the shape parameters $v$, $\delta_1$, and $\delta_2$, are gamma distributions that are independent of each other, with the following PDFs

$$\pi(v, \delta_1, \delta_2) \propto v^{a_1-1} \delta_1^{a_2-1} \delta_2^{a_3-1} e^{-(b_1 v + b_2 \delta_1 + b_3 \delta_2)}, \quad a_1, a_2, a_3, b_1, b_2, b_3 > 0, \quad (13)$$

where $a_1, b_1, a_2, b_2, a_3$, and $b_3$ are the hyper parameters. The class of gamma prior distributions is very adaptable because it may represent a wide range of previous information. To determine the hyper-parameters, we use INP and N-INP. In the case of INPs, the mean and variance of the designated gamma priors for $a_i$ and $b_i$ are equal to the mean and variance of the likelihood estimators for $\hat{v}, \hat{\delta}_1$ and $\hat{\delta}_2$. Consequently, we reach the result as stated in Dey et al. [32] by making the mean and variance of MLEs $\hat{v}, \hat{\delta}_1$ and $\hat{\delta}_2$ equal to those of the gamma priors.

$$\frac{1}{s} \sum_{j=1}^{s} \hat{\Psi}_i^j = \frac{a_i}{b_i}, \quad \frac{1}{s-1} \sum_{j=1}^{s} \left[ \hat{\Psi}_i^j - \frac{1}{s} \sum_{j=1}^{s} \hat{\Psi}_i^j \right]^2 = \frac{a_i}{b_i^2}, \quad i = 1, 2, 3, \quad \hat{\Psi} = (\hat{v}, \hat{\delta}_1, \hat{\delta}_2) ,$$

where $s$ is the iteration count of samples. Once the two equations above are solved, the estimated hyper-parameters are expressed as

$$a_i = \frac{\left[ s^{-1} \sum_{j=1}^{s} \hat{\Psi}_i^j \right]^2}{(s-1)^{-1} \sum_{j=1}^{s} \left[ \hat{\Psi}_i^j - \frac{1}{s} \sum_{j=1}^{s} \hat{\Psi}_i^j \right]^2}, \quad b_i = \frac{s^{-1} \sum_{j=1}^{s} \hat{\Psi}_i^j}{(s-1)^{-1} \sum_{j=1}^{s} \left[ \hat{\Psi}_i^j - \frac{1}{s} \sum_{j=1}^{s} \hat{\Psi}_i^j \right]^2}.$$

Moreover, in case of N-INP with hyper-parameters $a_i = b_i = 0.00001$; $i = 1, 2, 3$ (Congdon [33]) is included. As a result, the joint posterior distribution of $v$, $\delta_1$ and $\delta_2$ is determined by using the LF (7) and the joint prior distribution (13).

$$\Pi(v, \delta_1, \delta_2) = \frac{L(v, \delta_1, \delta_2) \pi(v, \delta_1, \delta_2)}{\int L(v, \delta_1, \delta_2) \pi(v, \delta_1, \delta_2) d(v, \delta_1, \delta_2)}. \quad (14)$$

The conditional posteriors are given as:

$$\Pi_1(v|\delta_1,\delta_2) \propto v^{a_1+m_1+m_2-1}e^{-\left(b_1v+\sum_{i_1=1}^{m_1}[v\ln q_{i_i}+[(\delta_1+1)+\delta_1(h_1(\mathscr{R}'_{i_1}+1)-1)]\ln(1+q_{i_{i_1}}^v)]\right)}$$

$$\times e^{-\sum_{i_2=1}^{m_2}[v\ln t_{i_2}+(\delta_2+1)\ln(1+t_{i_2}^{-v})-[(h_2(\mathscr{R}^*_{i_2}+1)-1]\ln[1-(1+t_{i_2}^{-v})^{-\delta_2}]]}\,,$$

$$\Pi_2(\delta_1|v,\delta_2) \propto \delta_1^{a_2+m_1-1}e^{-\left(b_2\delta_1+\sum_{i_1=1}^{m_1}[\delta_1+\delta_1(h_1(\mathscr{R}'_{i_1}+1)-1)]\ln(1+q_{i_{i_1}}^v)\right)}\,,$$

and

$$\Pi_3(\delta_2|v,\delta_1) \propto \delta_2^{a_3+m_2-1}e^{-\left(\delta_2 b_3+\sum_{i_2=1}^{m_2}[\delta_2\ln(1+t_{i_2}^{-v})-[(h_2(\mathscr{R}^*_{i_2}+1)-1]\ln[1-(1+t_{i_2}^{-v})^{-\delta_2}]]\right)}\,.$$

The Bayesian estimator of $\vartheta$ is defined as $\tilde{\vartheta}^{SE}$ and, $\tilde{\vartheta}^{LINx}$ respectively, where it minimizes the SE loss function, denoted as $L_{SE}(\vartheta,\tilde{\vartheta}^{SE})$, and LINx loss function, denoted as $L_{LINx}(\vartheta,\tilde{\vartheta}^{LINx})$,

$$L_{SE}(\vartheta,\tilde{\vartheta}^{SE}) = (\vartheta-\tilde{\vartheta}^{SE})^2,$$

$$L_{LINx}(\vartheta,\tilde{\vartheta}^{LINx}) = e^{\kappa(\vartheta-\tilde{\vartheta}^{LINx})} - \kappa(\vartheta-\tilde{\vartheta}^{LINx}) - 1,$$

and

$$\tilde{\vartheta}^{SE} = E(\vartheta), \quad \tilde{\vartheta}^{LINx} = \frac{-1}{\kappa}\ln[E(e^{-\kappa\vartheta})],$$

where $\kappa$ is an LINx scale parameter.

### 3.1 Metropolis-Hasting algorithm

The Metropolis-Hasting (MH) technique employs the following processes to extract a sample from the posterior density as specified by Eq (14)

**Step 1.** Initialize $\Psi$ with $\Psi = (v^{(0)},\delta_1^{(0)},\delta_2^{(0)}) = (\hat{v},\hat{\delta}_1,\hat{\delta}_2)$.

**Step 2.** For $i = 1, 2, \ldots, M$, carry out the subsequent procedures:

2.1: Set $\Psi = \Psi^{(i-1)}$.

2.2: Generate a new candidate parameter value $\Psi'$ by sampling from a normal distribution with a mean vector $\Psi^{(i-1)}$ and a small vector of standard deviations.

2.3: Calculate $\kappa = \frac{\pi^{\bullet\bullet}(\Psi')}{\pi^{\bullet\bullet}(\Psi)}$, where $\pi^{\bullet\bullet}(\cdot)$ is the posterior density in Eq (14).

2.4: Using the uniform distribution $U(0, 1)$, get a sample $u$ from the distribution.

2.5: Determine whether to accept or reject the new candidate $\Psi'$:

$$\Psi^{(i)} = \begin{cases} \Psi' & \text{if } u \leq \kappa, \\ \Psi & \text{otherwise} \end{cases}$$

Therefore, MCMC samples of $\vartheta$ are obtained as $\vartheta^{(i)}$, $i = 1, 2, \ldots, \tau$. Therefore, by replacing $\vartheta^{(i)}$ in Eq (5), $\vartheta$ may be calculated. Eventually, some of the original samples can be eliminated (burned in), and with the remaining samples, random samples of size $\tau$ taken from the posterior density can be used to compute BEs. Given SE and LINx, the BEs of a parametric function $\vartheta$ are as follows:

$$\hat{\vartheta}_{SE} = \frac{1}{\tau - l_B} \sum_{i=l_B}^{\tau} \vartheta^{(i)}, \tag{15}$$

and

$$\hat{\vartheta}_{LINx} = \frac{-1}{\kappa} \ln \left[ \frac{1}{\tau - l_B} \sum_{i=l_B}^{\tau} e^{-\kappa \vartheta^{(i)}} \right], \tag{16}$$

where the number of burn-in samples is denoted by $l_B$. The BEs of $\vartheta$ with regard to SE and LINx loss functions can be obtained by substituting $\vartheta^{(i)}$ in the aforementioned equations.

### 3.2 Highest posterior density

Employing the method described in [34], one can find the HPD intervals for the unknown parameters $\vartheta$ under POFIF-CS utilizing the samples acquired by the MH algorithm recommended in the previous paragraph. In this case, let $\vartheta^{(\rho)}$ be the $\rho$th quantile of $\vartheta$ is

$$\vartheta^{(\rho)} = \inf \{ \vartheta : \Pi(\vartheta \mid \mathbf{x}) \geq \rho \},$$

where $\Pi(\cdot)$ is the posterior function of $\vartheta$ and $0 < \rho < 1$. For a given $\vartheta^*$, it is possible to get a good estimate by simulating $\pi(\vartheta|\mathbf{x})$. This can be done as follows:

$$\Pi\{\vartheta \mid \mathbf{x}\} = \frac{1}{\tau - l_B} \sum_{i=l_B}^{\tau} I_{\vartheta \leq \vartheta^*}$$

Here $I_{\vartheta \leq \vartheta^*}$ is the indicator function. The proper estimate is then determined as

$$\hat{\Pi}(\vartheta^* \mid \mathbf{x}) = \begin{cases} 0 & \text{if } \vartheta^* < \vartheta_{l_B} \\ \sum_{j=l_B}^{i} \omega_j & \text{if } (\vartheta_i < \vartheta^* < \vartheta_{i+1} \\ 1 & \text{if } \vartheta^* > \vartheta_\tau \end{cases}$$

where $\omega_j = \frac{1}{\tau - l_B}$ and $\vartheta^j$ are the ordered values of $\vartheta^{(j)}$. Now, for $i = l_B, \ldots, \tau$, $\vartheta^{(\rho)}$ may be estimated by

$$\tilde{\vartheta}^{(\rho)} = \begin{cases} \vartheta_{l_B} & \text{if } \rho = 0 \\ \vartheta_i & \text{if } \sum_{j=l_B}^{i-1} \omega_j < \rho < \sum_{j=l_B}^{i} \omega_j. \end{cases}$$

Additionally, for $\vartheta$, let us compute a $100(1 - \rho)\%$ HPD credible interval as

$$HPD_j^\vartheta = \left( \tilde{\vartheta}^{\left(\frac{j}{\tau}\right)}, \quad \tilde{\vartheta}^{\left(\frac{j+(1-\rho)\tau}{\tau}\right)} \right)$$

for $j = l_B, \ldots, [\rho\tau]$, here $[a]$ represents indicates the largest integer that $\leq a$. Require selection of $HPD_{j^*}$ from a set of $HPD_j^\vartheta$s based on the smallest width.

## 4 Numerical results

The present section investigates the application of Monte Carlo simulation in evaluating the suggested estimations of the steady-state reliability parameter $\vartheta$ inside the framework of POFIF-CS. The main aim of this simulation study is to examine the characteristics and efficacy of the estimates generated by both ML and Bayesian methodologies. The R programming language was utilized for numerical computations, together with supplementary software tools, to facilitate equation solution and result extraction.

The simulation process involves 1000 replications. For the BXII($v, \delta_1$) and BIII($v, \delta_2$) distribution, four scenarios are considered: the first with $v = 0.5$, $\delta_1 = 0.25$, $\delta_2 = 0.5$; the second with $v = 0.5$, $\delta_1 = 0.25$, $\delta_2 = 0.5$; the third with $v = 1.5$, $\delta_1 = 1.25$, $\delta_2 = 1.5$; and the last with $v = 1.5$, $\delta_1 = 1.5$, $\delta_2 = 1.25$. The true value of stress-strength parameter $\vartheta$ is 0.8740, which close to one as high value of reliability, in the first two cases. On the other hand the true value of $\vartheta$ is 0.3405 in the last two cases, which is small value of reliability. Here, $v$ remains constant across both distributions, set at 0.5 and 1.5.

The number of groups in POFIF-CS ($h_1, h_2$) will be equal for each stress and will be denoted by $h$, where $h = 1$ and $h = 2$. Notably, when $h = 1$, the general POTII-CS is employed. Additionally, we will assume that the number of stages $m_1$ and $m_2$ are equal and the sample sizes $n_1$ and $n_2$ are different. The elimination of units from the life test is simulated using predefined values of $n_1$, $n_2$, $m_1$, and $m_2$, alongside various censoring patterns $\mathscr{R}'$ and $\mathscr{R}^*$ for each stress, as detailed in Table 1.

### 4.1 Required procedures in the Monte Carlo simulation

The following procedures are required for the simulation study.

**Table 1. Patterns of item removal for varying values of $m$ and $n$.**

| $(n_1, n_2)$ | $(m_1, m_2)$ | Censoring Scheme | | Scheme |
|---|---|---|---|---|
| | | $(\mathscr{R}'_1, \mathscr{R}'_2, \ldots, \mathscr{R}'_{m_1})$ | $(\mathscr{R}^*_1, \mathscr{R}^*_2, \ldots, \mathscr{R}^*_{m_2})$ | |
| (40, 40) | (30, 30) | $(10, 0^{*29})$ | $(10, 0^{*29})$ | $\mathscr{R}_1$ |
| | | $(0^{*29}, 10)$ | $(0^{*29}, 10)$ | $\mathscr{R}_2$ |
| | | $(1^{*10}, 0^{*20})$ | $(1^{*10}, 0^{*20})$ | $\mathscr{R}_3$ |
| | | $(0^{*20}, 1^{*10})$ | $(0^{*20}, 1^{*10})$ | $\mathscr{R}_4$ |
| (60, 50) | (40, 40) | $(20, 0^{*39})$ | $(10, 0^{*39})$ | $\mathscr{R}_5$ |
| | | $(0^{*39}, 20)$ | $(0^{*39}, 10)$ | $\mathscr{R}_6$ |
| | | $(1^{*20}, 0^{*20})$ | $(1^{*10}, 0^{*30})$ | $\mathscr{R}_7$ |
| | | $(0^{*20}, 1^{*20})$ | $(0^{*30}, 1^{*10})$ | $\mathscr{R}_8$ |
| (80, 100) | (60, 60) | $(20, 0^{*59})$ | $(40, 0^{*59})$ | $\mathscr{R}_9$ |
| | | $(0^{*59}, 20)$ | $(0^{*59}, 40)$ | $\mathscr{R}_{10}$ |
| | | $(1^{*20}, 0^{*40})$ | $(1^{*40}, 0^{*20})$ | $\mathscr{R}_{11}$ |
| | | $(0^{*40}, 1^{*20})$ | $(0^{*20}, 1^{*40})$ | $\mathscr{R}_{12}$ |

$(1^{(4)}, 2)$ indicate that the censoring scheme $(1, 1, 1, 1, 2)$.

**First Procedure:** Generate random POFIF-censored sample $Q_{i:m:n:h}$, by utilizing the provided $\mathscr{R}$, from the CDF $F(q)$ of the BXII($v, \delta_1$) distribution, considering it as POTII-censored sample from a population with CDF $1 - (1 - F(q))^h$ as in reference [35].

**Second Procedure:** Similarly, generate random POFIF-censored data $T_{i:m:n:h}$ from the CDF of BIII($v, \delta_2$), given $\mathscr{R}$.

**Third Procedure:** Calculate the MLEs for the parameters $v$, $\delta_1$, and $\delta_2$. Then, use these MLEs to determine the estimate for $\vartheta$ by inserting them into (5).

**Fourth Procedure:** Compute confidence intervals, both Asy-CIs and NA-CIs, for $\vartheta$ based on the variance calculation.

**Fifth Procedure:** Calculate the BE using the MH method in the following manner:

1. Let's examine two situations for prior distributions. In the first situation, employ the INP case to determine the values of hyper-parameters.

2. Take into account the second case that involves the N-INP, where all hyper-parameter values are set to 0.00001.

3. Employing MCMC and the MH method, generate 10,000 samples of $\vartheta$ from the posterior density, utilizing the hyper-parameters of the prior distributions that were given.

4. Remove the first 2,000 samples as burn-in from the whole set of 8,000 samples obtained by the posterior density.

5. Compute the BEs of $\vartheta$ utilizing two different loss functions: SE and LINx with $\kappa = 0.5$ for LINx-1 and $\kappa = -0.5$ for LINx-2, utilizing (15) and (16), respectively.

**Sixth Procedure:** Repeat from second procedure to the fifth procedure a total of 1,000 times and record all estimates.

**Seventh Procedure:** Compute two statistical metrics for point estimates: the mean estimate (Avg.) and the mean square error (MSE) of estimate via the subsequent formulas:

$$Avg.(\vartheta) = \frac{1}{1000}\sum_{l=1}^{1000}\hat{\vartheta}_l, \qquad \text{and} \qquad MSE(\vartheta) = \frac{1}{1000}\sum_{l=1}^{1000}(\hat{\vartheta}_l - \vartheta)^2.$$

Here, $\vartheta$ denotes the actual value of the steady-state with the provided parameters, while $\hat{\vartheta}$ represents the estimated value of the steady-state.

**Eighth Procedure:** Calculate statistical measures of performance for interval estimates: average interval length (AIL) and coverage probability (CP) in %.

## 4.2 Comments on outcomes

To get point estimates of $\vartheta$, We show the Avg. and MSE results for different $h$ values and the suggested censoring schemes $\mathscr{R}$, which are based on the process covered in the preceding subsection. Tables 2 and 3 show what happens when $v$ stays at 0.5 and ($\delta_1 = 0.25, \delta_2 = 0.5$) and ($\delta_1 = 0.5, \delta_2 = 0.25$), respectively. Additionally, Tables 4 and 5 correspond to cases where $v$ maintains a value of 1.5, with ($\delta_1 = 1.25, \delta_2 = 1.5$) and ($\delta_1 = 1.5, \delta_2 = 1.25$), respectively. In each table, the first row includes the Avg. and the second row includes the MSE of $\vartheta$. For interval estimation of $\vartheta$, we present four methods: Asy-CI, NA-CI, HPD in N-INP case, and HPD: INP case. The results of AIL and CP estimates for various values of $h$ and the proposed censoring

**Table 2. Avg. and MSE of the MLE and BEs for $\vartheta$ at $v = 0.5$, $\delta_1 = 0.25$, and $\delta_2 = 0.5$ under POFIF-CS ($h$, $m$, $n$).**

| $h$ | Scheme | | MLE | BE (MCMC): INP | | | BE (MCMC): N-INP | | |
|---|---|---|---|---|---|---|---|---|---|
| | | | | SE | LINx-1 | LINx-2 | SE | LINx-1 | LINx-2 |
| 1 | $\mathcal{R}_1$ | Avg. | 0.88251 | 0.91127 | 0.91121 | 0.91134 | 0.89023 | 0.88993 | 0.89023 |
| | | MSE | 0.00095 | 0.00146 | 0.00145 | 0.00146 | 0.00111 | 0.00111 | 0.00111 |
| | $\mathcal{R}_2$ | Avg. | 0.94451 | 0.92795 | 0.92789 | 0.92800 | 0.94872 | 0.94862 | 0.94872 |
| | | MSE | 0.00535 | 0.00295 | 0.00295 | 0.00296 | 0.00594 | 0.00592 | 0.00594 |
| | $\mathcal{R}_3$ | Avg. | 0.88818 | 0.91171 | 0.91164 | 0.91177 | 0.89603 | 0.89576 | 0.89603 |
| | | MSE | 0.00098 | 0.00149 | 0.00149 | 0.00150 | 0.00118 | 0.00117 | 0.00118 |
| | $\mathcal{R}_4$ | Avg. | 0.93756 | 0.92589 | 0.92584 | 0.92595 | 0.94201 | 0.94190 | 0.94201 |
| | | MSE | 0.00439 | 0.00273 | 0.00273 | 0.00274 | 0.00495 | 0.00494 | 0.00495 |
| | $\mathcal{R}_5$ | Avg. | 0.88104 | 0.81372 | 0.81000 | 0.81740 | 0.88706 | 0.88677 | 0.88706 |
| | | MSE | 0.00075 | 0.05416 | 0.05661 | 0.05176 | 0.00098 | 0.00100 | 0.00098 |
| | $\mathcal{R}_6$ | Avg. | 0.93823 | 0.92578 | 0.92565 | 0.92591 | 0.94161 | 0.94152 | 0.94161 |
| | | MSE | 0.00449 | 0.00301 | 0.00303 | 0.00299 | 0.00490 | 0.00489 | 0.00490 |
| | $\mathcal{R}_7$ | Avg. | 0.88492 | 0.85116 | 0.84835 | 0.85390 | 0.89082 | 0.89062 | 0.89082 |
| | | MSE | 0.00072 | 0.03164 | 0.03330 | 0.03006 | 0.00089 | 0.00089 | 0.00089 |
| | $\mathcal{R}_8$ | Avg. | 0.92933 | 0.91644 | 0.91606 | 0.91683 | 0.93345 | 0.93334 | 0.93345 |
| | | MSE | 0.00342 | 0.00535 | 0.00563 | 0.00508 | 0.00386 | 0.00385 | 0.00386 |
| | $\mathcal{R}_9$ | Avg. | 0.88068 | 0.74937 | 0.74242 | 0.75622 | 0.78094 | 0.77426 | 0.78094 |
| | | MSE | 0.00047 | 0.08807 | 0.09278 | 0.08352 | 0.07334 | 0.07781 | 0.07334 |
| | $\mathcal{R}_{10}$ | Avg. | 0.94968 | 0.83905 | 0.83441 | 0.84365 | 0.84882 | 0.84448 | 0.84882 |
| | | MSE | 0.00587 | 0.05327 | 0.05637 | 0.05026 | 0.04850 | 0.05101 | 0.04850 |
| | $\mathcal{R}_{11}$ | Avg. | 0.90759 | 0.77577 | 0.76912 | 0.78231 | 0.76642 | 0.75921 | 0.76642 |
| | | MSE | 0.00140 | 0.07312 | 0.07739 | 0.06900 | 0.07300 | 0.07804 | 0.07300 |
| | $\mathcal{R}_{12}$ | Avg. | 0.94013 | 0.80184 | 0.79563 | 0.80803 | 0.80918 | 0.80339 | 0.80918 |
| | | MSE | 0.00453 | 0.06972 | 0.07407 | 0.06547 | 0.06208 | 0.06615 | 0.06208 |
| 2 | $\mathcal{R}_1$ | Avg. | 0.93303 | 0.92471 | 0.92466 | 0.92476 | 0.93810 | 0.93800 | 0.93810 |
| | | MSE | 0.00378 | 0.00262 | 0.00261 | 0.00262 | 0.00438 | 0.00437 | 0.00438 |
| | $\mathcal{R}_2$ | Avg. | 0.94098 | 0.92875 | 0.92871 | 0.92880 | 0.94499 | 0.94492 | 0.94499 |
| | | MSE | 0.00469 | 0.00304 | 0.00304 | 0.00305 | 0.00523 | 0.00522 | 0.00523 |
| | $\mathcal{R}_3$ | Avg. | 0.93307 | 0.92466 | 0.92461 | 0.92471 | 0.93760 | 0.93750 | 0.93760 |
| | | MSE | 0.00376 | 0.00261 | 0.00261 | 0.00262 | 0.00429 | 0.00428 | 0.00429 |
| | $\mathcal{R}_4$ | Avg. | 0.94007 | 0.92807 | 0.92803 | 0.92812 | 0.94448 | 0.94441 | 0.94448 |
| | | MSE | 0.00458 | 0.00297 | 0.00296 | 0.00297 | 0.00516 | 0.00515 | 0.00516 |
| | $\mathcal{R}_5$ | Avg. | 0.93125 | 0.89463 | 0.89307 | 0.89624 | 0.93489 | 0.93481 | 0.93489 |
| | | MSE | 0.00351 | 0.01890 | 0.02003 | 0.01777 | 0.00392 | 0.00391 | 0.00392 |
| | $\mathcal{R}_6$ | Avg. | 0.93823 | 0.92787 | 0.92764 | 0.92808 | 0.94129 | 0.94123 | 0.94129 |
| | | MSE | 0.00428 | 0.00346 | 0.00353 | 0.00341 | 0.00468 | 0.00467 | 0.00468 |
| | $\mathcal{R}_7$ | Avg. | 0.92984 | 0.90482 | 0.90379 | 0.90582 | 0.93324 | 0.93317 | 0.93324 |
| | | MSE | 0.00329 | 0.01143 | 0.01204 | 0.01086 | 0.00367 | 0.00366 | 0.00367 |
| | $\mathcal{R}_8$ | Avg. | 0.93780 | 0.91786 | 0.91736 | 0.91835 | 0.94077 | 0.94072 | 0.94077 |
| | | MSE | 0.00423 | 0.00792 | 0.00817 | 0.00769 | 0.00461 | 0.00460 | 0.00461 |
| | $\mathcal{R}_9$ | Avg. | 0.93420 | 0.80556 | 0.79890 | 0.81210 | 0.81260 | 0.80726 | 0.81260 |
| | | MSE | 0.00376 | 0.06143 | 0.06578 | 0.05731 | 0.06434 | 0.06752 | 0.06434 |
| | $\mathcal{R}_{10}$ | Avg. | 0.94099 | 0.85867 | 0.85448 | 0.86274 | 0.82014 | 0.81430 | 0.82014 |
| | | MSE | 0.00456 | 0.04290 | 0.04494 | 0.04098 | 0.06047 | 0.06474 | 0.06047 |
| | $\mathcal{R}_{11}$ | Avg. | 0.93642 | 0.83467 | 0.82885 | 0.84032 | 0.78831 | 0.78247 | 0.78831 |
| | | MSE | 0.00402 | 0.04261 | 0.04585 | 0.03960 | 0.07354 | 0.07714 | 0.07354 |
| | $\mathcal{R}_{12}$ | Avg. | 0.94232 | 0.80370 | 0.79847 | 0.80893 | 0.81527 | 0.80946 | 0.81527 |
| | | MSE | 0.00474 | 0.06738 | 0.07093 | 0.06389 | 0.05775 | 0.06159 | 0.05775 |

**Table 3. Avg. and MSE of the MLE and BEs for $\vartheta$ at $v = 0.5$, $\delta_1 = 0.25$, and $\delta_2 = 0.5$ under POFIF-CS ($h$, $m$, $n$).**

| $h$ | Scheme | | MLE | BE (MCMC): INP | | | BE (MCMC): N-INP | | |
|---|---|---|---|---|---|---|---|---|---|
| | | | | SE | LINx-1 | LINx-2 | SE | LINx-1 | LINx-2 |
| 1 | $\mathcal{R}_1$ | Avg. | 0.88346 | 0.91347 | 0.91341 | 0.91353 | 0.89194 | 0.89165 | 0.89194 |
| | | MSE | 0.00094 | 0.00162 | 0.00161 | 0.00162 | 0.00109 | 0.00108 | 0.00109 |
| | $\mathcal{R}_2$ | Avg. | 0.94579 | 0.92991 | 0.92987 | 0.92996 | 0.95017 | 0.95008 | 0.95017 |
| | | MSE | 0.00565 | 0.00318 | 0.00317 | 0.00318 | 0.00622 | 0.00621 | 0.00622 |
| | $\mathcal{R}_3$ | Avg. | 0.88942 | 0.91449 | 0.91443 | 0.91455 | 0.89682 | 0.89656 | 0.89682 |
| | | MSE | 0.00111 | 0.00170 | 0.00170 | 0.00171 | 0.00131 | 0.00130 | 0.00131 |
| | $\mathcal{R}_4$ | Avg. | 0.93597 | 0.92715 | 0.92710 | 0.92720 | 0.94038 | 0.94026 | 0.94038 |
| | | MSE | 0.00430 | 0.00287 | 0.00287 | 0.00288 | 0.00484 | 0.00483 | 0.00484 |
| | $\mathcal{R}_5$ | Avg. | 0.88339 | 0.84182 | 0.83810 | 0.84542 | 0.88806 | 0.88766 | 0.88806 |
| | | MSE | 0.00088 | 0.03726 | 0.03925 | 0.03537 | 0.00141 | 0.00147 | 0.00141 |
| | $\mathcal{R}_6$ | Avg. | 0.93701 | 0.92206 | 0.92171 | 0.92240 | 0.94033 | 0.94024 | 0.94033 |
| | | MSE | 0.00437 | 0.00509 | 0.00523 | 0.00496 | 0.00478 | 0.00477 | 0.00478 |
| | $\mathcal{R}_7$ | Avg. | 0.88345 | 0.85341 | 0.85002 | 0.85672 | 0.88942 | 0.88922 | 0.88942 |
| | | MSE | 0.00072 | 0.03119 | 0.03322 | 0.02927 | 0.00082 | 0.00082 | 0.00082 |
| | $\mathcal{R}_8$ | Avg. | 0.93149 | 0.91390 | 0.91340 | 0.91440 | 0.93490 | 0.93480 | 0.93490 |
| | | MSE | 0.00362 | 0.01052 | 0.01092 | 0.01014 | 0.00402 | 0.00401 | 0.00402 |
| | $\mathcal{R}_9$ | Avg. | 0.88164 | 0.78918 | 0.78366 | 0.79461 | 0.76426 | 0.75744 | 0.76426 |
| | | MSE | 0.00043 | 0.06949 | 0.07301 | 0.06606 | 0.08131 | 0.08604 | 0.08131 |
| | $\mathcal{R}_{10}$ | Avg. | 0.94954 | 0.85296 | 0.84811 | 0.85780 | 0.83258 | 0.82730 | 0.83258 |
| | | MSE | 0.00583 | 0.04777 | 0.05105 | 0.04458 | 0.05619 | 0.05944 | 0.05619 |
| | $\mathcal{R}_{11}$ | Avg. | 0.90745 | 0.77647 | 0.77109 | 0.78182 | 0.77898 | 0.77368 | 0.77898 |
| | | MSE | 0.00136 | 0.08342 | 0.08718 | 0.07975 | 0.08433 | 0.08788 | 0.08433 |
| | $\mathcal{R}_{12}$ | Avg. | 0.94063 | 0.85027 | 0.84487 | 0.85555 | 0.80405 | 0.79804 | 0.80405 |
| | | MSE | 0.00462 | 0.04226 | 0.04534 | 0.03940 | 0.06667 | 0.07047 | 0.06667 |
| 2 | $\mathcal{R}_1$ | Avg. | 0.93492 | 0.92767 | 0.92763 | 0.92772 | 0.93926 | 0.93916 | 0.93926 |
| | | MSE | 0.00399 | 0.00292 | 0.00292 | 0.00293 | 0.00452 | 0.00451 | 0.00452 |
| | $\mathcal{R}_2$ | Avg. | 0.94183 | 0.93136 | 0.93132 | 0.93140 | 0.94579 | 0.94572 | 0.94579 |
| | | MSE | 0.00481 | 0.00333 | 0.00332 | 0.00333 | 0.00534 | 0.00533 | 0.00534 |
| | $\mathcal{R}_3$ | Avg. | 0.93346 | 0.92742 | 0.92737 | 0.92746 | 0.93797 | 0.93788 | 0.93797 |
| | | MSE | 0.00382 | 0.00290 | 0.00289 | 0.00290 | 0.00436 | 0.00435 | 0.00436 |
| | $\mathcal{R}_4$ | Avg. | 0.94050 | 0.93046 | 0.93042 | 0.93050 | 0.94454 | 0.94447 | 0.94454 |
| | | MSE | 0.00460 | 0.00322 | 0.00322 | 0.00322 | 0.00514 | 0.00513 | 0.00514 |
| | $\mathcal{R}_5$ | Avg. | 0.93161 | 0.89450 | 0.89233 | 0.89662 | 0.93559 | 0.93551 | 0.93559 |
| | | MSE | 0.00354 | 0.01743 | 0.01874 | 0.01619 | 0.00401 | 0.00401 | 0.00401 |
| | $\mathcal{R}_6$ | Avg. | 0.93876 | 0.91835 | 0.91756 | 0.91914 | 0.94184 | 0.94179 | 0.94184 |
| | | MSE | 0.00436 | 0.00756 | 0.00800 | 0.00713 | 0.00476 | 0.00475 | 0.00476 |
| | $\mathcal{R}_7$ | Avg. | 0.92940 | 0.91120 | 0.91006 | 0.91230 | 0.93310 | 0.93302 | 0.93310 |
| | | MSE | 0.00325 | 0.00923 | 0.00974 | 0.00878 | 0.00366 | 0.00365 | 0.00366 |
| | $\mathcal{R}_8$ | Avg. | 0.93760 | 0.90791 | 0.90655 | 0.90924 | 0.94077 | 0.94071 | 0.94077 |
| | | MSE | 0.00419 | 0.01108 | 0.01192 | 0.01030 | 0.00461 | 0.00460 | 0.00461 |
| | $\mathcal{R}_9$ | Avg. | 0.93375 | 0.82554 | 0.82094 | 0.83001 | 0.82127 | 0.81553 | 0.82127 |
| | | MSE | 0.00369 | 0.05308 | 0.05556 | 0.05069 | 0.05151 | 0.05497 | 0.05151 |
| | $\mathcal{R}_{10}$ | Avg. | 0.94133 | 0.86418 | 0.85913 | 0.86900 | 0.82656 | 0.82070 | 0.82656 |
| | | MSE | 0.00461 | 0.03392 | 0.03631 | 0.03172 | 0.04445 | 0.04808 | 0.04445 |
| | $\mathcal{R}_{11}$ | Avg. | 0.93764 | 0.83158 | 0.82602 | 0.83698 | 0.80933 | 0.80366 | 0.80933 |
| | | MSE | 0.00416 | 0.05139 | 0.05454 | 0.04841 | 0.06458 | 0.06822 | 0.06458 |
| | $\mathcal{R}_{12}$ | Avg. | 0.94167 | 0.79882 | 0.79281 | 0.80484 | 0.79865 | 0.79251 | 0.79865 |
| | | MSE | 0.00468 | 0.06974 | 0.07408 | 0.06547 | 0.07002 | 0.07415 | 0.07002 |

**Table 4. Avg. and MSE of the MLE and BEs for $\vartheta$ at $v = 1.5$, $\delta_1 = 1.5$, and $\delta_2 = 1.25$ under POFIF-CS ($h$, $m$, $n$).**

| $h$ | Scheme | | MLE | BE (MCMC): INP | | | BE (MCMC): N-INP | | |
|---|---|---|---|---|---|---|---|---|---|
| | | | | SE | LINx-1 | LINx-2 | SE | LINx-1 | LINx-2 |
| 1 | $\mathcal{R}_1$ | Avg. | 0.34853 | 0.33107 | 0.33070 | 0.33143 | 0.37677 | 0.37564 | 0.37677 |
| | | MSE | 0.00295 | 0.00052 | 0.00052 | 0.00051 | 0.00423 | 0.00415 | 0.00423 |
| | $\mathcal{R}_2$ | Avg. | 0.53598 | 0.40873 | 0.40834 | 0.40911 | 0.55958 | 0.55857 | 0.55958 |
| | | MSE | 0.04032 | 0.00501 | 0.00496 | 0.00507 | 0.05000 | 0.04957 | 0.05000 |
| | $\mathcal{R}_3$ | Avg. | 0.38161 | 0.34274 | 0.34236 | 0.34311 | 0.40974 | 0.40860 | 0.40974 |
| | | MSE | 0.00414 | 0.00039 | 0.00039 | 0.00039 | 0.00720 | 0.00704 | 0.00720 |
| | $\mathcal{R}_4$ | Avg. | 0.51290 | 0.39973 | 0.39934 | 0.40011 | 0.53726 | 0.53623 | 0.53726 |
| | | MSE | 0.03181 | 0.00386 | 0.00381 | 0.00391 | 0.04074 | 0.04034 | 0.04074 |
| | $\mathcal{R}_5$ | Avg. | 0.33858 | 0.32868 | 0.32835 | 0.32901 | 0.36029 | 0.35945 | 0.36029 |
| | | MSE | 0.00211 | 0.00058 | 0.00059 | 0.00057 | 0.00252 | 0.00249 | 0.00252 |
| | $\mathcal{R}_6$ | Avg. | 0.49809 | 0.40501 | 0.40467 | 0.40536 | 0.51622 | 0.51544 | 0.51622 |
| | | MSE | 0.02627 | 0.00450 | 0.00446 | 0.00455 | 0.03225 | 0.03198 | 0.03225 |
| | $\mathcal{R}_7$ | Avg. | 0.36383 | 0.33903 | 0.33869 | 0.33937 | 0.38534 | 0.38451 | 0.38534 |
| | | MSE | 0.00245 | 0.00040 | 0.00040 | 0.00040 | 0.00391 | 0.00384 | 0.00391 |
| | $\mathcal{R}_8$ | Avg. | 0.48062 | 0.39630 | 0.39596 | 0.39665 | 0.49919 | 0.49840 | 0.49919 |
| | | MSE | 0.02117 | 0.00346 | 0.00342 | 0.00350 | 0.02668 | 0.02643 | 0.02668 |
| | $\mathcal{R}_9$ | Avg. | 0.35273 | 0.33865 | 0.33838 | 0.33893 | 0.36681 | 0.36625 | 0.36681 |
| | | MSE | 0.00155 | 0.00042 | 0.00042 | 0.00042 | 0.00210 | 0.00207 | 0.00210 |
| | $\mathcal{R}_{10}$ | Avg. | 0.61007 | 0.49783 | 0.49755 | 0.49810 | 0.62000 | 0.61956 | 0.62000 |
| | | MSE | 0.07353 | 0.02505 | 0.02497 | 0.02514 | 0.07898 | 0.07874 | 0.07898 |
| | $\mathcal{R}_{11}$ | Avg. | 0.47892 | 0.41115 | 0.41086 | 0.41144 | 0.49232 | 0.49177 | 0.49232 |
| | | MSE | 0.02018 | 0.00532 | 0.00528 | 0.00536 | 0.02407 | 0.02390 | 0.02407 |
| | $\mathcal{R}_{12}$ | Avg. | 0.56738 | 0.46949 | 0.46921 | 0.46977 | 0.57849 | 0.57802 | 0.57849 |
| | | MSE | 0.05231 | 0.01693 | 0.01686 | 0.01700 | 0.05747 | 0.05724 | 0.05747 |
| 2 | $\mathcal{R}_1$ | Avg. | 0.56784 | 0.43338 | 0.43300 | 0.43376 | 0.59192 | 0.59104 | 0.59192 |
| | | MSE | 0.05334 | 0.00893 | 0.00886 | 0.00900 | 0.06482 | 0.06439 | 0.06482 |
| | $\mathcal{R}_2$ | Avg. | 0.60820 | 0.47373 | 0.47336 | 0.47410 | 0.62939 | 0.62859 | 0.62939 |
| | | MSE | 0.07353 | 0.01801 | 0.01792 | 0.01811 | 0.08523 | 0.08478 | 0.08523 |
| | $\mathcal{R}_3$ | Avg. | 0.58425 | 0.44419 | 0.44381 | 0.44456 | 0.60718 | 0.60634 | 0.60718 |
| | | MSE | 0.06070 | 0.01099 | 0.01092 | 0.01107 | 0.07230 | 0.07186 | 0.07230 |
| | $\mathcal{R}_4$ | Avg. | 0.60825 | 0.47004 | 0.46968 | 0.47041 | 0.63094 | 0.63016 | 0.63094 |
| | | MSE | 0.07341 | 0.01703 | 0.01694 | 0.01713 | 0.08593 | 0.08549 | 0.08593 |
| | $\mathcal{R}_5$ | Avg. | 0.56920 | 0.45084 | 0.45051 | 0.45118 | 0.58674 | 0.58607 | 0.58674 |
| | | MSE | 0.05356 | 0.01247 | 0.01240 | 0.01255 | 0.06186 | 0.06153 | 0.06186 |
| | $\mathcal{R}_6$ | Avg. | 0.59115 | 0.48570 | 0.48537 | 0.48603 | 0.60863 | 0.60798 | 0.60863 |
| | | MSE | 0.06425 | 0.02132 | 0.02122 | 0.02141 | 0.07321 | 0.07287 | 0.07321 |
| | $\mathcal{R}_7$ | Avg. | 0.56529 | 0.45217 | 0.45184 | 0.45251 | 0.58328 | 0.58261 | 0.58328 |
| | | MSE | 0.05155 | 0.01271 | 0.01263 | 0.01278 | 0.05992 | 0.05960 | 0.05992 |
| | $\mathcal{R}_8$ | Avg. | 0.59249 | 0.48235 | 0.48202 | 0.48268 | 0.61013 | 0.60950 | 0.61013 |
| | | MSE | 0.06485 | 0.02038 | 0.02029 | 0.02047 | 0.07397 | 0.07363 | 0.07397 |
| | $\mathcal{R}_9$ | Avg. | 0.57422 | 0.47946 | 0.47918 | 0.47973 | 0.58571 | 0.58526 | 0.58571 |
| | | MSE | 0.05539 | 0.01959 | 0.01951 | 0.01967 | 0.06087 | 0.06065 | 0.06087 |
| | $\mathcal{R}_{10}$ | Avg. | 0.64436 | 0.55480 | 0.55456 | 0.55504 | 0.65417 | 0.65381 | 0.65417 |
| | | MSE | 0.09313 | 0.04615 | 0.04604 | 0.04625 | 0.09916 | 0.09893 | 0.09916 |
| | $\mathcal{R}_{11}$ | Avg. | 0.62085 | 0.51951 | 0.51926 | 0.51977 | 0.63118 | 0.63079 | 0.63118 |
| | | MSE | 0.07931 | 0.03230 | 0.03221 | 0.03239 | 0.08518 | 0.08496 | 0.08518 |
| | $\mathcal{R}_{12}$ | Avg. | 0.63641 | 0.54281 | 0.54256 | 0.54305 | 0.64648 | 0.64611 | 0.64648 |
| | | MSE | 0.08835 | 0.04116 | 0.04106 | 0.04126 | 0.09439 | 0.09416 | 0.09439 |

**Table 5. Avg. and MSE of the MLE and BEs for $\vartheta$ at $v = 1.5$, $\delta_1 = 1.25$, and $\delta_2 = 1.5$ under POFIF-CS ($h, m, n$).**

| h | Scheme | | MLE | BE (MCMC): INP | | | BE (MCMC): N-INP | | |
|---|---|---|---|---|---|---|---|---|---|
| | | | | SE | LINx-1 | LINx-2 | SE | LINx-1 | LINx-2 |
| 1 | $\mathcal{R}_1$ | Avg. | 0.34363 | 0.32560 | 0.32527 | 0.32594 | 0.37152 | 0.37040 | 0.37152 |
| | | MSE | 0.00302 | 0.00062 | 0.00063 | 0.00061 | 0.00412 | 0.00404 | 0.00412 |
| | $\mathcal{R}_2$ | Avg. | 0.51662 | 0.39666 | 0.39630 | 0.39702 | 0.53972 | 0.53870 | 0.53972 |
| | | MSE | 0.03304 | 0.00346 | 0.00342 | 0.00350 | 0.04169 | 0.04129 | 0.04169 |
| | $\mathcal{R}_3$ | Avg. | 0.37660 | 0.33778 | 0.33743 | 0.33813 | 0.40413 | 0.40300 | 0.40413 |
| | | MSE | 0.00361 | 0.00033 | 0.00033 | 0.00033 | 0.00631 | 0.00617 | 0.00631 |
| | $\mathcal{R}_4$ | Avg. | 0.49860 | 0.38850 | 0.38814 | 0.38887 | 0.52215 | 0.52112 | 0.52215 |
| | | MSE | 0.02668 | 0.00254 | 0.00250 | 0.00257 | 0.03467 | 0.03429 | 0.03467 |
| | $\mathcal{R}_5$ | Avg. | 0.33993 | 0.32767 | 0.32736 | 0.32798 | 0.36070 | 0.35987 | 0.36070 |
| | | MSE | 0.00197 | 0.00052 | 0.00053 | 0.00051 | 0.00242 | 0.00238 | 0.00242 |
| | $\mathcal{R}_6$ | Avg. | 0.48204 | 0.39388 | 0.39355 | 0.39420 | 0.50084 | 0.50005 | 0.50084 |
| | | MSE | 0.02185 | 0.00323 | 0.00320 | 0.00327 | 0.02738 | 0.02712 | 0.02738 |
| | $\mathcal{R}_7$ | Avg. | 0.36092 | 0.33487 | 0.33456 | 0.33517 | 0.38111 | 0.38027 | 0.38111 |
| | | MSE | 0.00200 | 0.00034 | 0.00035 | 0.00034 | 0.00332 | 0.00325 | 0.00332 |
| | $\mathcal{R}_8$ | Avg. | 0.47474 | 0.38962 | 0.38930 | 0.38995 | 0.49404 | 0.49324 | 0.49404 |
| | | MSE | 0.01957 | 0.00272 | 0.00269 | 0.00275 | 0.02513 | 0.02488 | 0.02513 |
| | $\mathcal{R}_9$ | Avg. | 0.34899 | 0.33449 | 0.33423 | 0.33475 | 0.36313 | 0.36257 | 0.36313 |
| | | MSE | 0.00150 | 0.00044 | 0.00044 | 0.00044 | 0.00195 | 0.00192 | 0.00195 |
| | $\mathcal{R}_{10}$ | Avg. | 0.58449 | 0.47765 | 0.47738 | 0.47792 | 0.59513 | 0.59468 | 0.59513 |
| | | MSE | 0.06032 | 0.01905 | 0.01898 | 0.01912 | 0.06562 | 0.06539 | 0.06562 |
| | $\mathcal{R}_{11}$ | Avg. | 0.46132 | 0.39787 | 0.39759 | 0.39814 | 0.47425 | 0.47370 | 0.47425 |
| | | MSE | 0.01551 | 0.00355 | 0.00352 | 0.00359 | 0.01878 | 0.01863 | 0.01878 |
| | $\mathcal{R}_{12}$ | Avg. | 0.54780 | 0.45354 | 0.45327 | 0.45381 | 0.55842 | 0.55793 | 0.55842 |
| | | MSE | 0.04373 | 0.01302 | 0.01295 | 0.01308 | 0.04821 | 0.04800 | 0.04821 |
| 2 | $\mathcal{R}_1$ | Avg. | 0.54341 | 0.41651 | 0.41615 | 0.41687 | 0.56617 | 0.56523 | 0.56617 |
| | | MSE | 0.04286 | 0.00603 | 0.00598 | 0.00609 | 0.05256 | 0.05215 | 0.05256 |
| | $\mathcal{R}_2$ | Avg. | 1.05940 | 0.44981 | 0.44946 | 0.45016 | 0.59376 | 0.59288 | 0.59376 |
| | | MSE | 3.28185 | 0.01217 | 0.01209 | 0.01225 | 0.06614 | 0.06570 | 0.06614 |
| | $\mathcal{R}_3$ | Avg. | 0.55777 | 0.42578 | 0.42542 | 0.42614 | 0.58134 | 0.58044 | 0.58134 |
| | | MSE | 0.04857 | 0.00748 | 0.00742 | 0.00754 | 0.05929 | 0.05886 | 0.05929 |
| | $\mathcal{R}_4$ | Avg. | 0.81518 | 0.44809 | 0.44773 | 0.44845 | 0.60129 | 0.60044 | 0.60129 |
| | | MSE | 1.56196 | 0.01175 | 0.01167 | 0.01182 | 0.06947 | 0.06903 | 0.06947 |
| | $\mathcal{R}_5$ | Avg. | 0.54438 | 0.43364 | 0.43332 | 0.43397 | 0.56186 | 0.56115 | 0.56186 |
| | | MSE | 0.04274 | 0.00891 | 0.00885 | 0.00897 | 0.05010 | 0.04979 | 0.05010 |
| | $\mathcal{R}_6$ | Avg. | 0.55227 | 0.46271 | 0.46239 | 0.46302 | 0.57260 | 0.57190 | 0.57260 |
| | | MSE | 0.05360 | 0.01509 | 0.01502 | 0.01517 | 0.05511 | 0.05479 | 0.05511 |
| | $\mathcal{R}_7$ | Avg. | 0.54637 | 0.43807 | 0.43775 | 0.43839 | 0.56494 | 0.56425 | 0.56494 |
| | | MSE | 0.04363 | 0.00976 | 0.00970 | 0.00982 | 0.05160 | 0.05129 | 0.05160 |
| | $\mathcal{R}_8$ | Avg. | 0.57502 | 0.46049 | 0.46018 | 0.46081 | 0.57860 | 0.57792 | 0.57860 |
| | | MSE | 0.14134 | 0.01457 | 0.01450 | 0.01465 | 0.05788 | 0.05755 | 0.05788 |
| | $\mathcal{R}_9$ | Avg. | 0.54549 | 0.45837 | 0.45811 | 0.45864 | 0.55674 | 0.55627 | 0.55674 |
| | | MSE | 0.04288 | 0.01415 | 0.01408 | 0.01421 | 0.04759 | 0.04739 | 0.04759 |
| | $\mathcal{R}_{10}$ | Avg. | 0.53157 | 0.52014 | 0.51989 | 0.52040 | 0.60180 | 0.60137 | 0.60180 |
| | | MSE | 0.06687 | 0.03242 | 0.03233 | 0.03251 | 0.06913 | 0.06891 | 0.06913 |
| | $\mathcal{R}_{11}$ | Avg. | 0.59067 | 0.49500 | 0.49474 | 0.49526 | 0.60167 | 0.60125 | 0.60167 |
| | | MSE | 0.06320 | 0.02405 | 0.02397 | 0.02413 | 0.06880 | 0.06858 | 0.06880 |
| | $\mathcal{R}_{12}$ | Avg. | 0.52069 | 0.51421 | 0.51397 | 0.51446 | 0.60438 | 0.60396 | 0.60438 |
| | | MSE | 0.07783 | 0.03031 | 0.03023 | 0.03040 | 0.07021 | 0.06999 | 0.07021 |

**Table 6. AILs and CPs for $\vartheta$ at $v = 0.5$, $\delta_1 = 0.25$, and $\delta_2 = 0.5$ under POFIF-CS $(h, m, n)$.**

| $h$ | Scheme | Asy-CI | | NA-CI | | HPD: INP | | HPD: N-INP | |
|---|---|---|---|---|---|---|---|---|---|
| | | AIL | CP | AIL | CP | AIL | CP | AIL | CP |
| 1 | $\mathscr{R}_1$ | 0.12296 | 98.8 | 0.12306 | 99.2 | 0.03402 | 98.4 | 0.11691 | 99.2 |
| | $\mathscr{R}_2$ | 0.08172 | 98.1 | 0.08175 | 97.7 | 0.03401 | 96.0 | 0.07273 | 98.0 |
| | $\mathscr{R}_3$ | 0.10691 | 98.4 | 0.10697 | 98.4 | 0.03184 | 98.8 | 0.10458 | 98.8 |
| | $\mathscr{R}_4$ | 0.07681 | 99.2 | 0.07683 | 99.2 | 0.02761 | 96.0 | 0.07722 | 98.4 |
| | $\mathscr{R}_5$ | 0.09542 | 98.0 | 0.09546 | 98.8 | 0.77289 | 98.2 | 0.09109 | 97.6 |
| | $\mathscr{R}_6$ | 0.07399 | 99.2 | 0.07401 | 99.6 | 0.03589 | 96.0 | 0.07432 | 99.6 |
| | $\mathscr{R}_7$ | 0.09973 | 98.4 | 0.09978 | 98.4 | 0.48715 | 98.2 | 0.09611 | 98.0 |
| | $\mathscr{R}_8$ | 0.07049 | 99.2 | 0.07051 | 99.2 | 0.07192 | 97.6 | 0.06502 | 98.8 |
| | $\mathscr{R}_9$ | 0.08224 | 98.0 | 0.08227 | 98.4 | 0.72296 | 98.2 | 0.80307 | 96.9 |
| | $\mathscr{R}_{10}$ | 0.04593 | 98.4 | 0.04593 | 98.4 | 0.33188 | 98.2 | 0.34443 | 96.9 |
| | $\mathscr{R}_{11}$ | 0.06738 | 98.0 | 0.06739 | 98.0 | 0.60060 | 98.2 | 0.52263 | 96.9 |
| | $\mathscr{R}_{12}$ | 0.05338 | 98.4 | 0.05339 | 98.4 | 0.39705 | 98.2 | 0.39383 | 96.9 |
| 2 | $\mathscr{R}_1$ | 0.07121 | 98.1 | 0.07123 | 97.7 | 0.02844 | 98.8 | 0.06927 | 96.8 |
| | $\mathscr{R}_2$ | 0.05726 | 99.6 | 0.05727 | 99.6 | 0.02755 | 97.2 | 0.05493 | 98.4 |
| | $\mathscr{R}_3$ | 0.07078 | 99.6 | 0.07079 | 99.6 | 0.02800 | 98.4 | 0.06537 | 99.2 |
| | $\mathscr{R}_4$ | 0.05816 | 99.6 | 0.05816 | 99.6 | 0.02489 | 97.2 | 0.06036 | 98.8 |
| | $\mathscr{R}_5$ | 0.05221 | 98.0 | 0.05222 | 98.0 | 0.24149 | 98.2 | 0.05272 | 98.0 |
| | $\mathscr{R}_6$ | 0.04901 | 98.8 | 0.04902 | 98.8 | 0.07971 | 98.4 | 0.05111 | 99.2 |
| | $\mathscr{R}_7$ | 0.05556 | 98.8 | 0.05556 | 98.8 | 0.17989 | 98.2 | 0.05576 | 98.4 |
| | $\mathscr{R}_8$ | 0.05187 | 99.2 | 0.05187 | 99.2 | 0.08495 | 99.2 | 0.04892 | 98.0 |
| | $\mathscr{R}_9$ | 0.04849 | 99.2 | 0.04849 | 99.2 | 0.34009 | 98.2 | 0.32961 | 96.9 |
| | $\mathscr{R}_{10}$ | 0.03459 | 99.1 | 0.03459 | 99.1 | 0.22207 | 98.2 | 0.22076 | 96.9 |
| | $\mathscr{R}_{11}$ | 0.03965 | 99.6 | 0.03965 | 99.6 | 0.35003 | 98.2 | 0.34809 | 96.9 |
| | $\mathscr{R}_{12}$ | 0.03569 | 97.8 | 0.03569 | 97.8 | 0.32188 | 98.2 | 0.32999 | 96.9 |

schemes $\mathscr{R}_i$ are also provided. Tables 6 and 7 correspond to cases where $v$ maintains a value of 0.5, with $(\delta_1 = 0.25, \delta_2 = 0.5)$ and $(\delta_1 = 0.5, \delta_2 = 0.25)$, respectively. Additionally, Tables 8 and 9 correspond to cases where $v$ maintains a value of 1.5, with $(\delta_1 = 1.25, \delta_2 = 1.5)$ and $(\delta_1 = 1.5, \delta_2 = 1.25)$, respectively. From the results in Tables 2 to 9, we can draw some observations:

- It's evident that as the sample size ($n$ and $m$) increases, the MSE decreases, and likewise, the Avg. approach the true value of the parameter $\vartheta$ for both MLEs and BE methods.

- The INP case tends to outperform the N-INP case. When comparing loss function methods (SE, LINx-1, LINx-2) in Bayesian analysis, no single method demonstrates superior efficiency over others.

- For interval estimation, there's a noticeable decrease in the AIL with an increase in the size of the monitored samples. Regarding the number of groups in the POFIF-CS, an increase in $h$ leads to an increase in MSE for all methods. Similarly, the CP ranges from 95% to 99%, with improvements observed in the INP case compared to the N-INP case for Bayesian methods. In contrast, among non-Bayesian methods, the Asy-CI exhibits better efficiency than the NA-CI. The ranking of interval estimation methods, in terms of AILs, can be as follows:

$$\text{HPD : INP} \leq \text{HPD : N}-\text{INP} \leq \text{Asy}-\text{CI} \leq \text{NA}-\text{CI}$$

**Table 7. AILs and CPs for $\vartheta$ at $v = 0.5$, $\delta_1 = 0.5$, and $\delta_2 = 0.25$ under POFIF-CS $(h, m, n)$.**

| $h$ | Scheme | Asy-CI | | NA-CI | | HPD: INP | | HPD: N-INP | |
|---|---|---|---|---|---|---|---|---|---|
| | | AIL | CP | AIL | CP | AIL | CP | AIL | CP |
| 1 | $\mathscr{R}_1$ | 0.11221 | 99.0 | 0.11229 | 99.0 | 0.84940 | 98.2 | 0.10799 | 97.5 |
| | $\mathscr{R}_2$ | 0.08651 | 99.5 | 0.08654 | 99.8 | 0.59677 | 98.2 | 0.08254 | 98.3 |
| | $\mathscr{R}_3$ | 0.11215 | 99.3 | 0.11222 | 99.8 | 0.72221 | 98.2 | 0.09909 | 99.3 |
| | $\mathscr{R}_4$ | 0.08530 | 98.8 | 0.08533 | 99.0 | 0.61684 | 98.2 | 0.08587 | 97.5 |
| | $\mathscr{R}_5$ | 0.09715 | 98.5 | 0.09720 | 98.5 | 0.76801 | 98.2 | 0.13701 | 97.8 |
| | $\mathscr{R}_6$ | 0.07131 | 99.0 | 0.07133 | 99.5 | 0.58787 | 98.2 | 0.08722 | 98.0 |
| | $\mathscr{R}_7$ | 0.09122 | 98.8 | 0.09126 | 99.0 | 0.68322 | 98.2 | 0.12121 | 98.0 |
| | $\mathscr{R}_8$ | 0.06967 | 98.0 | 0.06969 | 98.5 | 0.67258 | 98.2 | 0.11732 | 99.2 |
| | $\mathscr{R}_9$ | 0.07811 | 98.4 | 0.07814 | 99.0 | 0.66402 | 98.2 | 0.69719 | 96.9 |
| | $\mathscr{R}_{10}$ | 0.07037 | 98.1 | 0.07039 | 97.7 | 0.70075 | 98.2 | 0.42567 | 96.9 |
| | $\mathscr{R}_{11}$ | 0.06650 | 98.0 | 0.06652 | 98.4 | 0.32158 | 97.8 | 0.06719 | 98.4 |
| | $\mathscr{R}_{12}$ | 0.05261 | 98.8 | 0.05261 | 98.8 | 0.42077 | 98.3 | 0.05823 | 98.0 |
| 2 | $\mathscr{R}_1$ | 0.06337 | 99.3 | 0.06339 | 99.3 | 0.54412 | 98.2 | 0.07043 | 98.5 |
| | $\mathscr{R}_2$ | 0.04709 | 98.0 | 0.04710 | 98.0 | 0.45763 | 98.2 | 0.05541 | 97.0 |
| | $\mathscr{R}_3$ | 0.05895 | 99.5 | 0.05896 | 99.5 | 0.52465 | 98.2 | 0.06185 | 98.5 |
| | $\mathscr{R}_4$ | 0.05435 | 99.2 | 0.05435 | 99.2 | 0.49042 | 98.2 | 0.05881 | 98.5 |
| | $\mathscr{R}_5$ | 0.05393 | 98.2 | 0.05394 | 98.2 | 0.43745 | 98.2 | 0.43421 | 96.9 |
| | $\mathscr{R}_6$ | 0.03958 | 98.8 | 0.03958 | 98.8 | 0.37251 | 98.2 | 0.37206 | 96.9 |
| | $\mathscr{R}_7$ | 0.04879 | 99.4 | 0.04879 | 99.4 | 0.53938 | 98.2 | 0.40520 | 96.9 |
| | $\mathscr{R}_8$ | 0.04360 | 98.2 | 0.04360 | 98.2 | 0.44837 | 98.2 | 0.37714 | 96.9 |
| | $\mathscr{R}_9$ | 0.04846 | 99.2 | 0.04847 | 99.2 | 0.43148 | 97.4 | 0.05676 | 98.8 |
| | $\mathscr{R}_{10}$ | 0.04066 | 98.1 | 0.04067 | 98.1 | 0.37940 | 96.9 | 0.09486 | 97.5 |
| | $\mathscr{R}_{11}$ | 0.04579 | 98.8 | 0.04580 | 99.2 | 0.42087 | 97.8 | 0.05213 | 98.4 |
| | $\mathscr{R}_{12}$ | 0.03526 | 98.1 | 0.03526 | 98.6 | 0.37979 | 98.6 | 0.09058 | 99.5 |

- The convergence of MCMC estimates using the MH algorithm can be visualized through scatter plots and histograms for the estimated parameter $\vartheta$, as illustrated in Figs 1 and 2. These visualizations are based on the scheme $\mathscr{R}_1$, given $h = 2$ for BXII($v = 0.5$, $\delta_1 = 0.5$) and BIII($v$, $\delta_2 = 0.25$) distributions, where INP and N-INP are employed. These graphical representations demonstrate the normality of the generated posterior samples and provide insight into the convergence behavior of the Bayesian estimators. Notably, these graphs illustrate how the BEs converge towards the true parameter values and highlight the efficiency of the INP over the N-INP case.

## 5 Modeling to real data

Here, we examine two real datasets to demonstrate the implementation of our suggested estimation methods. The datasets provided in this study comprise the breakdown periods of insulating fluid between electrodes, which were recorded at different voltage levels [36]. The failure times (in minutes) for the insulating fluid between two electrodes under the influence of 36 kV ($Q$) and 34 kV ($T$) are presented in Table 10.

The BXII($v$, $\delta_1$) and BIII($v$, $\delta_2$) distributions are initially applied independently to datasets $Q$ and $T$. First and foremost, it's crucial to ascertain the suitability of each distribution for analyzing its respective dataset. Based on the estimated parameters the Kolmogorov-Smirnov

**Table 8. AILs and CPs for $\vartheta$ at $\nu = 1.5$, $\delta_1 = 1.5$, and $\delta_2 = 1.25$ under POFIF-CS ($h$, $m$, $n$).**

| $h$ | Scheme | Asy-CI | | NA-CI | | HPD: INP | | HPD: N-INP | |
|---|---|---|---|---|---|---|---|---|---|
| | | AIL | CP | AIL | CP | AIL | CP | AIL | CP |
| 1 | $\mathscr{R}_1$ | 0.21480 | 98.5 | 0.21828 | 99.3 | 0.18396 | 98.5 | 0.20397 | 98.5 |
| | $\mathscr{R}_2$ | 0.18193 | 97.8 | 0.18287 | 98.0 | 0.15512 | 97.5 | 0.18247 | 96.0 |
| | $\mathscr{R}_3$ | 0.17894 | 96.8 | 0.18059 | 98.5 | 0.16065 | 98.0 | 0.18316 | 96.8 |
| | $\mathscr{R}_4$ | 0.17921 | 99.0 | 0.18018 | 99.3 | 0.15250 | 98.0 | 0.17202 | 98.0 |
| | $\mathscr{R}_5$ | 0.17982 | 97.8 | 0.18191 | 98.0 | 0.15919 | 97.8 | 0.16535 | 97.5 |
| | $\mathscr{R}_6$ | 0.16028 | 97.3 | 0.16101 | 98.5 | 0.13663 | 97.0 | 0.15184 | 97.8 |
| | $\mathscr{R}_7$ | 0.17048 | 97.5 | 0.17209 | 98.5 | 0.15645 | 97.5 | 0.16616 | 97.3 |
| | $\mathscr{R}_8$ | 0.15308 | 97.5 | 0.15373 | 98.3 | 0.13419 | 96.8 | 0.14932 | 97.0 |
| | $\mathscr{R}_9$ | 0.13405 | 97.8 | 0.13491 | 98.3 | 0.11975 | 96.3 | 0.13115 | 98.0 |
| | $\mathscr{R}_{10}$ | 0.12233 | 98.0 | 0.12255 | 98.5 | 0.10771 | 97.0 | 0.12154 | 97.5 |
| | $\mathscr{R}_{11}$ | 0.12405 | 98.3 | 0.12442 | 99.3 | 0.10972 | 98.5 | 0.12326 | 99.8 |
| | $\mathscr{R}_{12}$ | 0.12017 | 97.3 | 0.12042 | 98.3 | 0.11472 | 98.3 | 0.11721 | 98.5 |
| 2 | $\mathscr{R}_1$ | 0.15103 | 98.0 | 0.15151 | 99.0 | 0.12684 | 97.5 | 0.15828 | 99.0 |
| | $\mathscr{R}_2$ | 0.36843 | 98.1 | 0.19490 | 96.3 | 0.13512 | 98.9 | 0.19182 | 98.9 |
| | $\mathscr{R}_3$ | 0.14456 | 98.0 | 0.14497 | 98.3 | 0.12739 | 97.5 | 0.14394 | 97.0 |
| | $\mathscr{R}_4$ | 0.37962 | 98.1 | 0.00000 | 95.8 | 0.11881 | 97.6 | 0.15239 | 98.0 |
| | $\mathscr{R}_5$ | 0.13396 | 97.3 | 0.13430 | 98.5 | 0.12525 | 98.8 | 0.12948 | 98.5 |
| | $\mathscr{R}_6$ | 0.39551 | 98.1 | 0.00000 | 97.1 | 0.11544 | 97.7 | 0.14308 | 97.4 |
| | $\mathscr{R}_7$ | 0.13065 | 98.8 | 0.13096 | 96.0 | 0.11406 | 97.8 | 0.12182 | 98.8 |
| | $\mathscr{R}_8$ | 0.13898 | 98.1 | 0.13876 | 94.5 | 0.11189 | 97.1 | 0.13793 | 99.1 |
| | $\mathscr{R}_9$ | 0.10497 | 97.5 | 0.10513 | 98.3 | 0.09492 | 98.0 | 0.10316 | 96.8 |
| | $\mathscr{R}_{10}$ | 0.12468 | 98.1 | 0.12876 | 96.5 | 0.11068 | 95.7 | 0.12871 | 97.9 |
| | $\mathscr{R}_{11}$ | 0.10803 | 98.1 | 0.10123 | 97.7 | 0.08638 | 97.5 | 0.09401 | 97.2 |
| | $\mathscr{R}_{12}$ | 0.10774 | 98.1 | 0.11384 | 97.0 | 0.09245 | 97.3 | 0.11323 | 98.6 |

distance for $Q$ is 0.1721 and the corresponding p-value is 0.7045, the Kolmogorov-Smirnov distance for $T$ is 0.1235 and the corresponding p-value is 0.9003. The p-values suggest that the he BXII($\nu$, $\delta_1$) and BIII($\nu$, $\delta_2$) give an acceptable fit for these data sets. The empirical distribution functions for the data set $Q$ and $T$, are given in Fig 3.

Based on the complete data set, the MLEs $\hat{\nu}, \hat{\delta}_1, \hat{\delta}_2$ are (2.4589, 0.3766, 3.0670) and the MLE of $\vartheta$ is 0.5394. The BEs of $\vartheta$ with N-INP via MCMC method is 0.5583. The length of 95% Asy-CI, NA-CI, and HPD credible interval for $\vartheta$ are 0.3225, 0.3402, and 0.1016 respectively. For illustrative purposes, two different POFIF-CS have been generated from the above data sets (Table 11):

Based on the POFIF-C data from two real datasets, we computed point and interval estimates for the reliability of SS $\vartheta$. Since prior information on unknown parameters is unavailable for BEs, we employed the N-INP approach. Utilizing SE and LINx loss functions (with $\kappa = 0.5$ for LINx-1 and $\kappa = -0.5$ for LINx-2), BEs of $\vartheta$ were computed using the MCMC method and utilizing MH algorithm. A Markov chain with 10,000 observations was generated, discarding the initial 2,000 observations as 'burn-in'. The MLE for $\vartheta$ was determined to be 0.8354. Additionally, BEs of $\vartheta$ were computed as 0.8453, 0.8449, and 0.8456 under SE, LINx-1, and LINx-2 loss functions, respectively. The 95% Asy-CI was calculated as (0.6786, 0.9921), while the NA-CI was (0.6924, 1.0077). Lastly, the HPD credible interval was determined to be (0.7738, 0.91023). From Fig 4, it is evident that the MCMC chain converges very well.

**Table 9. AILs and CPs for $\vartheta$ at $v$ = 1.5, $\delta_1$ = 1.25, and $\delta_2$ = 1.5 under POFIF-CS ($h, m, n$).**

| $h$ | Scheme | Asy-CI | | NA-CI | | HPD: INP | | HPD: N-INP | |
|---|---|---|---|---|---|---|---|---|---|
| | | AIL | CP | AIL | CP | AIL | CP | AIL | CP |
| 1 | $\mathscr{R}_1$ | 0.21653 | 97.6 | 0.22007 | 98.8 | 0.07381 | 97.6 | 0.21796 | 96.8 |
| | $\mathscr{R}_2$ | 0.17245 | 97.6 | 0.17321 | 99.2 | 0.06234 | 96.4 | 0.17271 | 99.2 |
| | $\mathscr{R}_3$ | 0.19537 | 96.0 | 0.19744 | 98.0 | 0.06786 | 96.4 | 0.18391 | 96.8 |
| | $\mathscr{R}_4$ | 0.18297 | 97.2 | 0.18394 | 98.0 | 0.07175 | 96.8 | 0.19145 | 96.4 |
| | $\mathscr{R}_5$ | 0.17538 | 96.8 | 0.17734 | 97.6 | 0.06684 | 97.6 | 0.16874 | 96.8 |
| | $\mathscr{R}_6$ | 0.15063 | 97.6 | 0.15120 | 98.0 | 0.06291 | 97.2 | 0.14328 | 98.0 |
| | $\mathscr{R}_7$ | 0.16498 | 98.0 | 0.16640 | 98.0 | 0.07016 | 98.8 | 0.16494 | 97.2 |
| | $\mathscr{R}_8$ | 0.15121 | 98.4 | 0.15183 | 98.8 | 0.06510 | 99.2 | 0.13484 | 98.4 |
| | $\mathscr{R}_9$ | 0.14676 | 97.6 | 0.14786 | 97.6 | 0.07097 | 97.2 | 0.13829 | 96.8 |
| | $\mathscr{R}_{10}$ | 0.11868 | 98.0 | 0.11887 | 98.4 | 0.06094 | 96.8 | 0.11264 | 96.8 |
| | $\mathscr{R}_{11}$ | 0.12107 | 98.8 | 0.12139 | 99.2 | 0.06299 | 98.8 | 0.11041 | 97.2 |
| | $\mathscr{R}_{12}$ | 0.11868 | 99.2 | 0.11889 | 99.2 | 0.06078 | 97.2 | 0.10901 | 96.4 |
| 2 | $\mathscr{R}_1$ | 0.14550 | 99.2 | 0.14590 | 99.2 | 0.05168 | 96.8 | 0.13766 | 98.0 |
| | $\mathscr{R}_2$ | 0.16401 | 98.4 | 0.16450 | 99.6 | 0.05964 | 96.4 | 0.16389 | 96.8 |
| | $\mathscr{R}_3$ | 0.14308 | 99.2 | 0.14344 | 99.2 | 0.04995 | 98.0 | 0.13262 | 98.8 |
| | $\mathscr{R}_4$ | 0.16175 | 99.2 | 0.16222 | 99.2 | 0.05837 | 99.6 | 0.15829 | 98.8 |
| | $\mathscr{R}_5$ | 0.13164 | 97.6 | 0.13193 | 98.0 | 0.06309 | 97.2 | 0.13163 | 98.8 |
| | $\mathscr{R}_6$ | 0.15273 | 98.4 | 0.15316 | 98.8 | 0.06217 | 97.2 | 0.14142 | 99.2 |
| | $\mathscr{R}_7$ | 0.13346 | 98.4 | 0.13377 | 98.0 | 0.05854 | 97.2 | 0.12475 | 97.6 |
| | $\mathscr{R}_8$ | 0.12662 | 99.2 | 0.12685 | 99.2 | 0.05389 | 98.4 | 0.12163 | 98.8 |
| | $\mathscr{R}_9$ | 0.11056 | 98.0 | 0.11073 | 96.8 | 0.05443 | 97.6 | 0.10417 | 97.6 |
| | $\mathscr{R}_{10}$ | 0.11283 | 98.8 | 0.11297 | 97.6 | 0.05385 | 97.2 | 0.10341 | 99.6 |
| | $\mathscr{R}_{11}$ | 0.08853 | 98.8 | 0.08861 | 98.8 | 0.04830 | 99.2 | 0.08659 | 96.4 |
| | $\mathscr{R}_{12}$ | 0.10319 | 98.0 | 0.10330 | 98.4 | 0.05263 | 98.4 | 0.10451 | 98.4 |

# 6 Conclusion

The inference of $\vartheta = P[T < Q]$ is investigated in this paper with progressively first-failure censored data, applying both Bayesian and non-Bayesian methods. Assume that the strength ($Q$), which follow the BXII distribution is independent of stress ($T$), which follow the BIII distribution. For classical method, the MLE, Asy-CI, and NA-CI of $\vartheta = P[T < Q]$ are produced. The Bayes estimator of $\vartheta$ under squared error and linear exponential loss functions are generated by means of non-informative and gamma informative priors. To get Bayes estimators and the corresponding credible intervals, it is proposed to employ Markov chain Monte Carlo methods for Bayesian estimation. A Monte Carlo numerical analysis is performed to assess the performance of the various estimators developed based on mean squared error, average interval length and coverage probability. We conclude from the simulation research that for both estimation techniques, the average of estimates approaches the real value of the $\vartheta$ and the MSEs decline as sample sizes rise. Overall, the Bayesian estimate in the INP situation often performs better. For both techniques, the MSE increases with an increase in number of groups $h$ in the POFIF-CS. The Asy-CI performs more efficiently than NA-CI. The AILs based on Bayesian technique are smaller than corresponding under classical method. Lastly, an analytical application to real data is explored for demonstrative purpose.

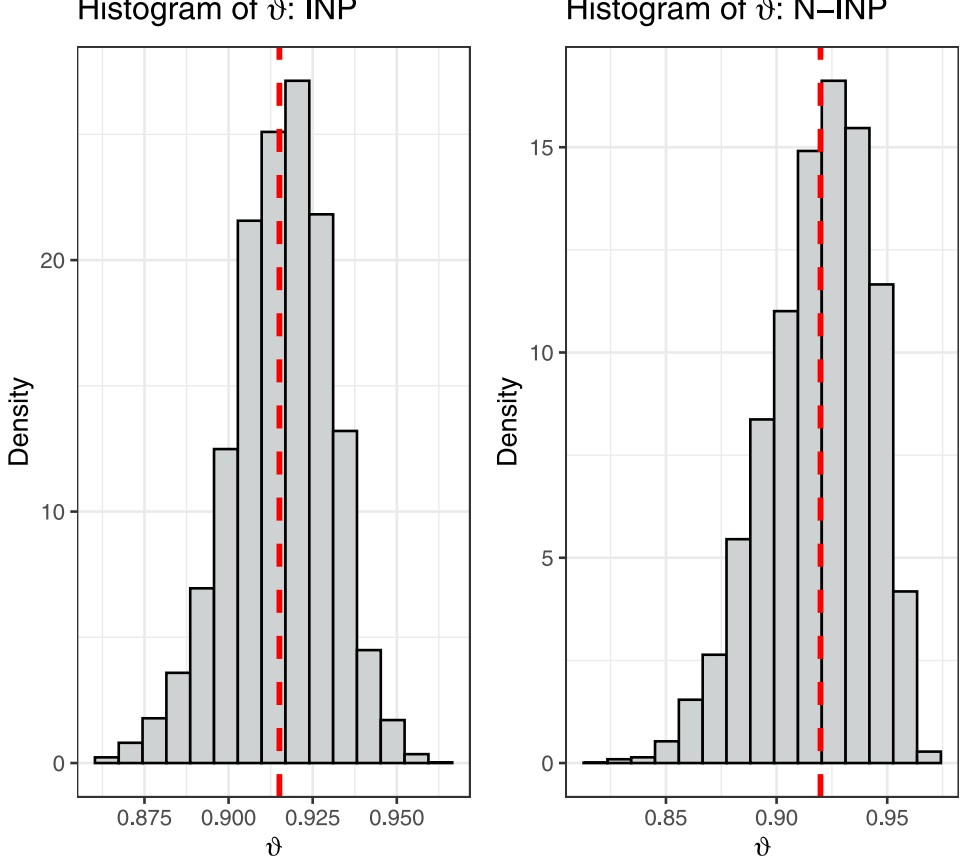

**Fig 1. Trace plot of MCMC estimates of ϑ.**

## Appendix 1

The second-order partial derivative are provided via

$$\frac{\partial^2 L^{\bullet\bullet}}{\partial v^2} = -\frac{m_1 + m_2}{v^2} - (\delta_1 + 1)\sum_{i_1=1}^{m_1} \frac{q_{i_1}^{-v}[\ln(q_{i_1})]^2}{(1 + q_{i_1}^{-v})^2} - \sum_{i_1=1}^{m_1} \frac{[h_1(R'_{i_1} + 1) - 1]\delta_1 q_{i_1}^{-v}[\ln(q_{i_1})]^2}{(1 + q_{i_1}^{-v})^2}$$

$$+ (\delta_2 + 1)\sum_{i_2=1}^{m_2} \frac{t_{i_2}^{v}[\ln(t_{i_2})]^2}{(1 + t_{i_2}^{v})^2} - \sum_{i_2=1}^{m_2} \frac{\delta_2(\delta_2 + 1)[h_2(R^*_{i_2} + 1) - 1](1 + t_{i_2}^{-v})^{-\delta_2 - 2}t_{i_2}^{-2v}[\ln(t_{i_2})]^2}{[1 - (1 + t_{i_2}^{-v})^{-\delta_2}]}$$

$$+ \sum_{i_2=1}^{m_2} \frac{\delta_2[h_2(R^*_{i_2} + 1) - 1]t_{i_2}^{-v}(1 + t_{i_2}^{-v})^{-\delta_2 - 1}[\ln(t_{i_2})]^2}{[1 - (1 + t_{i_2}^{-v})^{-\delta_2}]}$$

$$- \sum_{i_2=1}^{m_2} \frac{\delta_2^{2}[h_2(R^*_{i_2} + 1) - 1][\ln(t_{i_2})]^2(1 + t_{i_2}^{-v})^{-2\delta_2 - 2}t_{i_2}^{-2v}}{[1 - (1 + t_{i_2}^{-v})^{-\delta_2}]^2},$$

$$\frac{\partial^2 L^{\bullet\bullet}}{\partial \delta_1^{2}} = -\frac{m_1}{\delta_1^{2}},$$

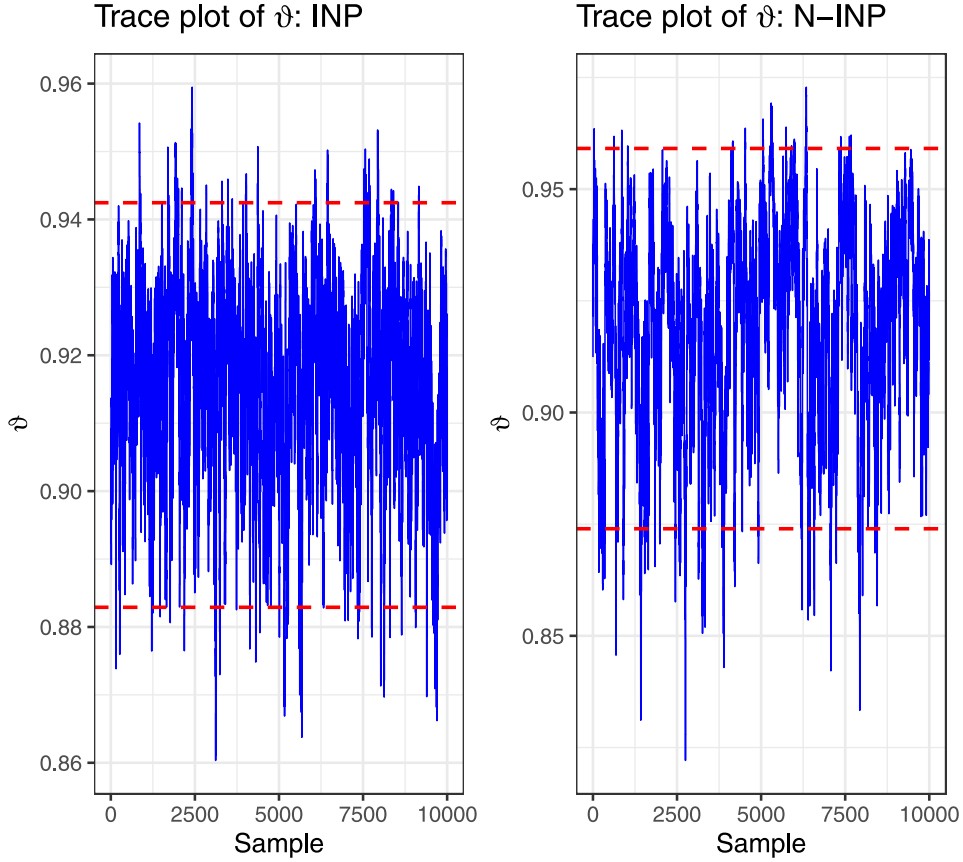

**Fig 2. Histogram of MCMC estimates of $\vartheta$.**

$$\frac{\partial^2 L^{\bullet\bullet}}{\partial \delta_2^2} = -\frac{m_2}{\delta_2^2} - \sum_{i_2=1}^{m_2} \frac{\left[h_2\left(R_{i_2}^* + 1\right) - 1\right]\left(1 + t_{i_2}^{-\nu}\right)^{\delta_2}\left[\ln\left(1 + t_{i_2}^{-\nu}\right)\right]^2}{\left[\left(1 + t_{i_2}^{-\nu}\right)^{\delta_2} - 1\right]^2},$$

$$\frac{\partial^2 L^{\bullet\bullet}}{\partial \delta_1 \partial \delta_2} = 0,$$

$$\frac{\partial^2 L^{\bullet\bullet}}{\partial \delta_1 \partial \nu} = -\sum_{i_1=1}^{m_1} \frac{q_{i_1}^{\nu} \ln\left(q_{i_1}\right)}{1 + q_{i_1}^{\nu}} - \sum_{i_1=1}^{m_1} \frac{\left[h_1\left(R'_{i_1} + 1\right) - 1\right]q_{i_1}^{\nu} \ln\left(q_{i_1}\right)}{1 + q_{i_1}^{\nu}},$$

**Table 10. Two datasets.**

| Q | 0.35 | 0.59 | 0.96 | 0.99 | 1.69 | 1.97 | 2.07 | 2.58 | 2.71 | 2.90 |
|---|------|------|------|------|------|------|------|------|------|------|
|   | 3.67 | 3.99 | 5.35 | 13.77 | 25.50 |  |  |  |  |  |
| T | 0.19 | 0.78 | 0.96 | 1.31 | 2.78 | 3.16 | 4.15 | 4.67 | 4.85 | 6.50 |
|   | 7.35 | 8.01 | 8.27 | 12.06 | 31.75 | 32.52 | 33.91 | 36.71 | 72.89 |  |

(a)

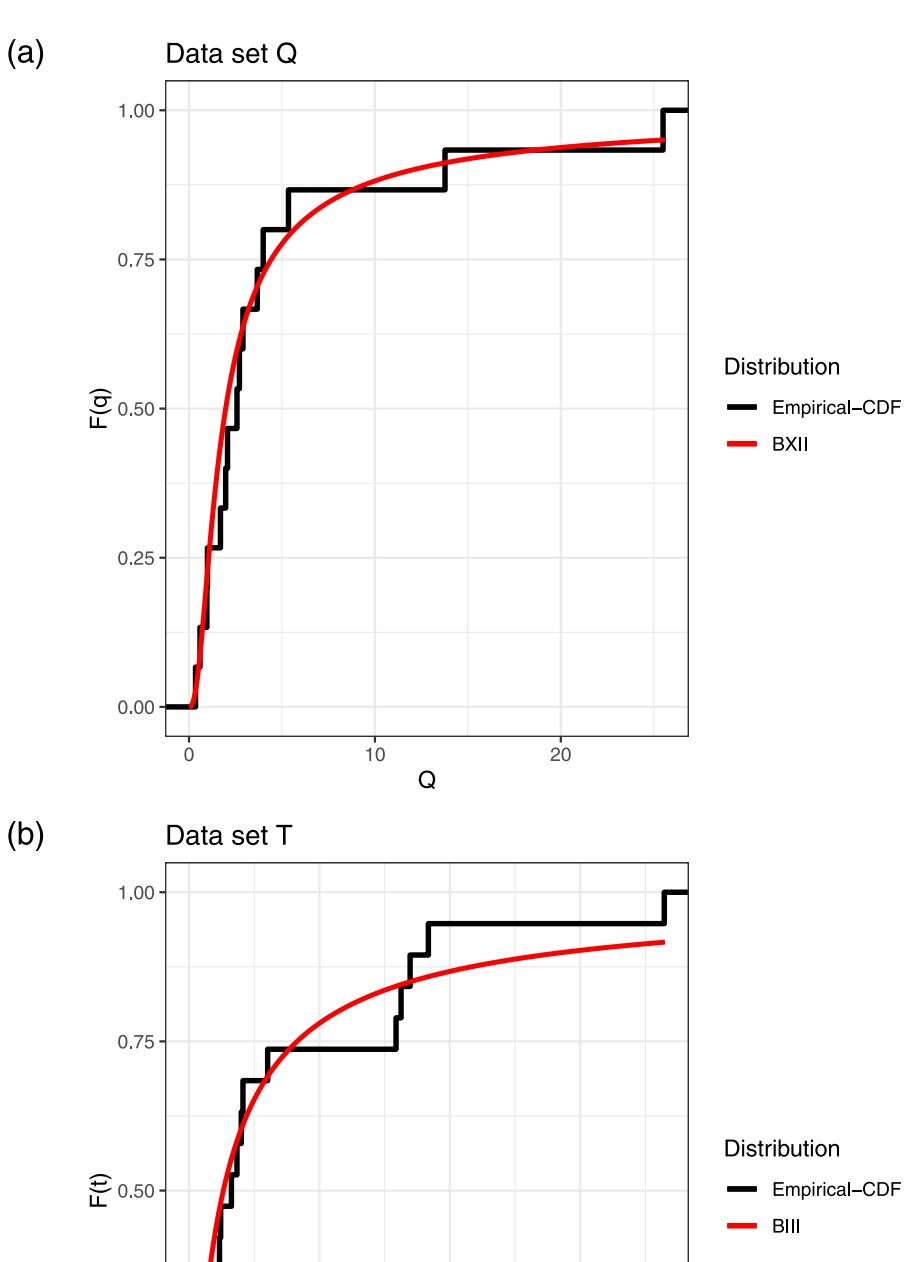

(b)

**Fig 3. The empirical distribution function and fitted distribution functions for data sets *Q* and *T*.**

**Table 11. The POFIF-C samples from the real data sets.**

| Set 1 | $(h_1, n_1, m_1) = (2, 15, 10)$ | $\mathscr{R}' = (5, 0^{*9})$ | 0.35, 0.99, 1.69, 1.97, 2.58,2.71, |
|---|---|---|---|
| | | | 2.90, 3.67, 5.35, 13.77 |
| Set 2 | $(h_2, n_2, m_2) = (2, 19, 15)$ | $\mathscr{R}^* = (4, 0^{*15})$ | 0.19, 0.78, 0.96, 2.78, 3.16, 4.67, |
| | | | 6.50, 7.35, 8.01, 8.27, 12.06, 31.75, |
| | | | 32.52, 33.91, 36.71 |

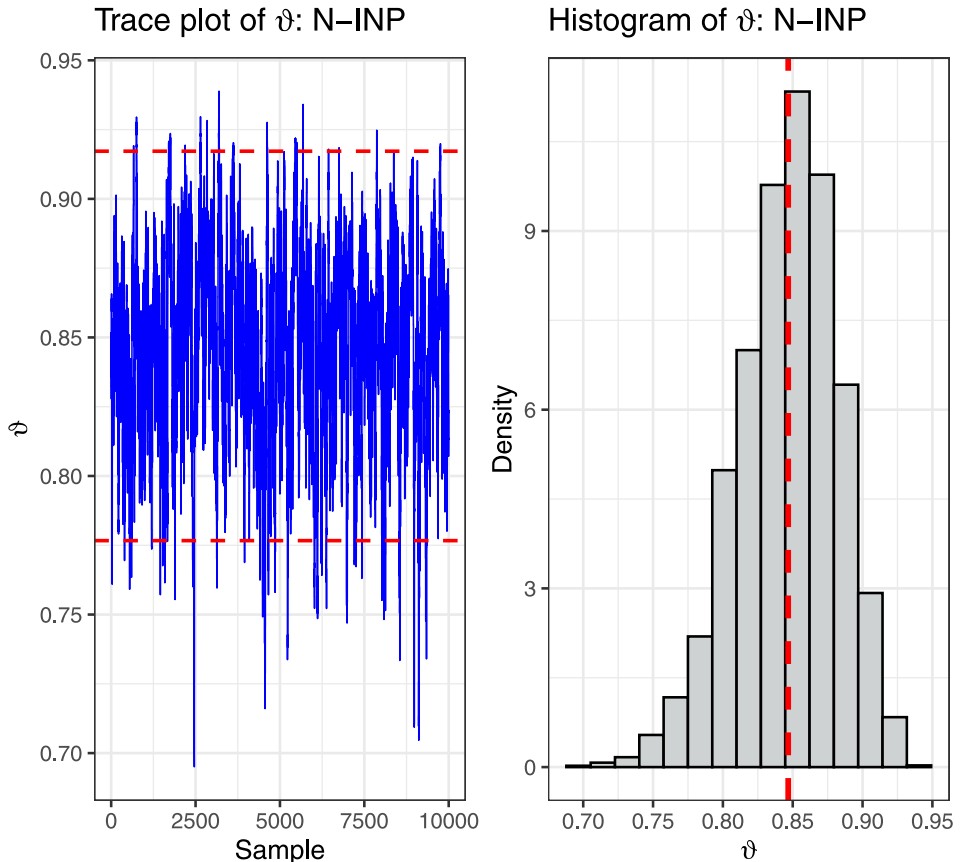

**Fig 4. Trace plot and histogram of MCMC samples using MH algorithm.**

and

$$\frac{\partial^2 L^{\bullet\bullet}}{\partial \delta_2 \partial v} = \sum_{i_2=1}^{m_2} \frac{\ln(t_{i_2})}{1+t_{i_2}^v} + \sum_{i_2=1}^{m_2} \frac{[h_2(R_{i_2}^*+1)-1]\ln(t_{i_2})}{[(1+t_{i_2}^{-v})^{\hat{\delta}_2}-1][1+t_{i_2}^v]} + \delta_2 \sum_{i_2=1}^{m_2} \frac{[h_2(R_{i_2}^*+1)-1]\ln(t_{i_2})\ln(1+t_{i_2}^{-v})}{t_{i_2}^v[(1+t_{i_2}^{-v})^{\delta_2}-1]^2(1+t_{i_2}^{-v})^{1-\delta_2}}.$$

## Author Contributions

**Conceptualization:** Salem A. Alyami, Amal S. Hassan, Ibrahim Elbatal, Olayan Albalawi, Mohammed Elgarhy, Ahmed R. El-Saeed.

**Formal analysis:** Salem A. Alyami, Amal S. Hassan, Ibrahim Elbatal, Olayan Albalawi, Mohammed Elgarhy, Ahmed R. El-Saeed.

**Methodology:** Salem A. Alyami, Mohammed Elgarhy, Ahmed R. El-Saeed.

**Software:** Salem A. Alyami, Amal S. Hassan, Ibrahim Elbatal, Olayan Albalawi, Mohammed Elgarhy.

**Writing – original draft:** Salem A. Alyami, Amal S. Hassan, Ibrahim Elbatal, Olayan Albalawi, Mohammed Elgarhy, Ahmed R. El-Saeed.

**Writing – review & editing:** Salem A. Alyami, Amal S. Hassan, Ibrahim Elbatal, Mohammed Elgarhy, Ahmed R. El-Saeed.

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
