## [Decision Letter · Decision Letter 0]

23 Jul 2024

PONE-D-24-22736Bayesian and Non-Bayesian Analysis for Stress-Strength Model Based on  Progressively First Failure Censoring with ApplicationsPLOS ONE

Dear Dr. Elgarhy,

Thank you for submitting your manuscript to PLOS ONE. After careful consideration, we feel that it has merit but does not fully meet PLOS ONE’s publication criteria as it currently stands. Therefore, we invite you to submit a revised version of the manuscript that addresses the points raised during the review process.

We look forward to receiving your revised manuscript.

Kind regards,

Mudassar Rashid, Ph.D

Academic Editor

PLOS ONE

Journal Requirements:

 [This work was supported and funded by the Deanship of Scientific Research at Imam Mohammad Ibn Saud Islamic University (IMSIU) (grant number IMSIU-RPP2023003).].  

4. Please update your submission to use the PLOS LaTeX template. The template and more information on our requirements for LaTeX submissions can be found at http://journals.plos.org/plosone/s/latex.

5. We note that your Data Availability Statement is currently as follows: [All relevant data are within the manuscript and its Supporting Information files.]

Reviewers' comments:

Reviewer's Responses to Questions

**Comments to the Author**

1. Is the manuscript technically sound, and do the data support the conclusions?

Reviewer #1: Yes

Reviewer #2: Yes

2. Has the statistical analysis been performed appropriately and rigorously? 

Reviewer #1: Yes

Reviewer #2: Yes

3. Have the authors made all data underlying the findings in their manuscript fully available?

Reviewer #1: Yes

Reviewer #2: Yes

4. Is the manuscript presented in an intelligible fashion and written in standard English?

Reviewer #1: Yes

Reviewer #2: Yes

5. Review Comments to the Author

Reviewer #1: 1. The breadth of topics covered, from censoring techniques to the application of SS models in various fields, demonstrates a thorough understanding of the subject matter. This provides a solid background for readers new to the topic while offering depth for those familiar with the field. The combination of classical and Bayesian estimation approaches, along with the use of MCMC, promises to offer significant advancements in the field.

2. The method leverages the well-established theory of MLE to derive parameter estimates. The use of the observed Fisher information matrix provides a robust framework for assessing the variance and constructing confidence intervals.

3. however, the iterative nature of the Newton-Raphson method can be computationally intensive, especially for large datasets. The normal approximation for constructing CIs may not be accurate for small sample sizes or for parameters that are near the boundary of the parameter space.

4. Bayesian estimation allows for the incorporation of prior information, which can be particularly useful when dealing with limited data. The use of the Metropolis-Hasting algorithm facilitates the sampling from complex posterior distributions.

5. However, The choice of priors can significantly influence the results, and inappropriate priors can lead to biased estimates. The computational complexity of the Metropolis-Hasting algorithm can be high, particularly for large datasets or complex models.

6. The paper lacks a detailed explanation of the choice of specific parameter values (,δ1,δ2). Providing context or rationale behind these choices would strengthen the study’s relevance.

7. Assuming equal sample sizes (n1, n2) and stages (m1, m2) simplifies the simulation but may not reflect real-world scenarios. Investigating unequal sample sizes and stages could offer a more comprehensive understanding.

8. While MSE and Avr. are standard performance metrics, additional metrics like bias or variance could provide a deeper analysis.

9. The study mentions that INP performs better overall but does not delve into the reasons behind this trend. A deeper exploration of why INP outperforms N-INP would enhance the findings’ interpretability.

Reviewer #2: The manuscript is a very interesting and written in the standard format of a theoretical article. Congratulations to authors for making a valuale contribution and the manuscript is in a perfect shape. The manuscript may be accpeted in its current form.

6. PLOS authors have the option to publish the peer review history of their article (what does this mean?). If published, this will include your full peer review and any attached files.

Reviewer #1: **Yes: **Dr. Shahid Akbar

Reviewer #2: **Yes: **Asad Ul Islam Khan

---

## [Author Response · Author response to Decision Letter 0]

16 Aug 2024

Response to the reviewer’s comments 

Submission ID PONE-D-24-22736

Bayesian and Non-Bayesian Analysis for Stress-Strength Model Based on Progressively First Failure Censoring with Applications

PLOS ONE

Dear Editors and Reviewers: 

 First, the authors would like to thank the Editor in Chief, Associate editor, Lead editor, Academic editor and Anonymous referees for spending their time on the manuscript carefully. The comments of the editors and reviewers are valuable. We have taken all the suggestions/comments positively and did our best to incorporate all these suggestions in the revised version. Our point wise responses to the reviewer’s comments/suggestions are given below. 

Response to Editor 

 Dear Sir/Mam, we appreciate your time in handling our paper and providing suggestions for improvement. We believe the quality of the revised version has considerably improved and hope that you find the revised manuscript satisfactory this time.

Reply to the Reviewer #1 

 The breadth of topics covered, from censoring techniques to the application of SS models in various fields, demonstrates a thorough understanding of the subject matter. This provides a solid background for readers new to the topic while offering depth for those familiar with the field. The combination of classical and Bayesian estimation approaches, along with the use of MCMC, promises to offer significant advancements in the field.

 Answer: we appreciate your feedback, many thanks for your positive comment.

 The method leverages the well-established theory of MLE to derive parameter estimates. The use of the observed Fisher information matrix provides a robust framework for assessing the variance and constructing confidence intervals.

Answer: we appreciate your feedback, many thanks for your positive comment.

 however, the iterative nature of the Newton-Raphson method can be computationally intensive, especially for large datasets. The normal approximation for constructing CIs may not be accurate for small sample sizes or for parameters that are near the boundary of the parameter space.

 Answer: we appreciate your feedback, we remove the sample size n=20 and m=15 from the article and replace them with another two large sets as seen in Table 1. Indeed, sample sizes and patterns of POFIF-CS were selected to ensure that the number of stages (m1 and m2) for each type of stress is sufficient to form CIs. The results of simulation study for old and new samples are provided in Tables 2 to 9. 

 Bayesian estimation allows for the incorporation of prior information, which can be particularly useful when dealing with limited data. The use of the Metropolis-Hasting algorithm facilitates the sampling from complex posterior distributions.

Answer: we appreciate your feedback, many thanks for your positive comment.

 However, the choice of priors can significantly influence the results, and inappropriate priors can lead to biased estimates. The computational complexity of the Metropolis-Hasting algorithm can be high, particularly for large datasets or complex models.

Answer: we appreciate your feedback, we agree with this fact, so in this study a convergence diagnostic of the Metropolis-Hastings algorithm has been assessed to ensure reliable results as provided in Figure 1. These diagnostics included trace plots, and histogram. The results, presented in Figure 1, indicate satisfactory convergence of the Markov chains."

 The paper lacks a detailed explanation of the choice of specific parameter values (v,δ1,δ2). Providing context or rationale behind these choices would strengthen the study’s relevance.

Answer: we appreciate your feedback, the selection of the assumed parameters allows for estimating the stress strength with values ranging from zero to one. Indeed, for the first and second case the true value of stress strength parameter ϑ=0.874, which is close to one, while in the third and fourth case the true value of stress strength parameter ϑ=0.3405, which is small value of reliability. Also, this issue is explained in simulation study.

 Assuming equal sample sizes (n1, n2) and stages (m1, m2) simplifies the simulation but may not reflect real-world scenarios. Investigating unequal sample sizes and stages could offer a more comprehensive understanding.

Answer: we appreciate your feedback, new patterns of POFIF-CS were selected and defined, including differences in sample sizes while keeping the number of stages equal. This resulted in a total of 12 patterns as showed in Table (1). Additionally, two cases for the number of groups h were specified. It should be noted that this required revising the practical part in the attached table for the new sample sizes.

 While MSE and Avg. are standard performance metrics, additional metrics like bias or variance could provide a deeper analysis.

Answer: we appreciate your feedback, we agree with you Dear Prof. on this point, but the research may expand to include a number of tables equal to twice the current results. Moreover, selecting a measure for the mean (Avg.) can be considered equivalent to addressing bias, and choosing a measure for the MSE can be considered equivalent to addressing variance. We have tried to balance between the mean and the variance of the obtained results Also, the MSE is a crucial metric for evaluating the performance of a model or estimator. It quantifies the average squared difference between the predicted (or estimated) values and the actual observed values. It is defined by MSE = variance + (bias)2. In a future study, we will take this point into consideration.

 The study mentions that INP performs better overall but does not delve into the reasons behind this trend. A deeper exploration of why INP outperforms N-INP would enhance the findings’ interpretability.

Answer: we appreciate your feedback, but dear Prof. as shown in the tables of results and the convergence plots for the MCMC study, in both informative and non-informative cases, it became clear that the results in the informative case were better. Therefore, we recommend using the informative case for estimation. This does not preclude the possibility of better informative cases, but determining which cases are better requires a sensitivity analysis. However, we were able to obtain a more efficient estimate that serves the current purpose. The main reasons for this are primarily twofold: the prior distribution assumed in this study was a Gamma distribution, and the second reason is the method of determining the hyper-parameter values (elicitation), which were derived from a previous study. We note and emphasize once again that there is still the potential for better informative cases, but this requires effort beyond the scope of this research, which is not the aim of this study.

Reply to the Reviewer #2 

 Reviewer #2: The manuscript is a very interesting and written in the standard format of a theoretical article. Congratulations to authors for making a value contribution and the manuscript is in a perfect shape. The manuscript may be accepted in its current form.

 Answer: Answer: we appreciate your feedback, many thanks for your positive report.

---

## [Decision Letter · Decision Letter 1]

16 Oct 2024

Bayesian and Non-Bayesian Analysis for Stress-Strength Model Based on  Progressively First Failure Censoring with Applications

PONE-D-24-22736R1

Dear Dr. Elgarhy,

We’re pleased to inform you that your manuscript has been judged scientifically suitable for publication and will be formally accepted for publication once it meets all outstanding technical requirements.

Kind regards,

Inés P. Mariño, Ph.D.

Academic Editor

PLOS ONE

Additional Editor Comments (optional):

Reviewers' comments:

Reviewer's Responses to Questions

**Comments to the Author**

1. If the authors have adequately addressed your comments raised in a previous round of review and you feel that this manuscript is now acceptable for publication, you may indicate that here to bypass the “Comments to the Author” section, enter your conflict of interest statement in the “Confidential to Editor” section, and submit your "Accept" recommendation.

Reviewer #2: All comments have been addressed

2. Is the manuscript technically sound, and do the data support the conclusions?

Reviewer #2: Yes

3. Has the statistical analysis been performed appropriately and rigorously? 

Reviewer #2: Yes

4. Have the authors made all data underlying the findings in their manuscript fully available?

Reviewer #2: Yes

5. Is the manuscript presented in an intelligible fashion and written in standard English?

Reviewer #2: Yes

6. Review Comments to the Author

Reviewer #2: There are no further comments, so the manuscript may be accepted as it is. The authors have incorporated all the comments from the First round review.

7. PLOS authors have the option to publish the peer review history of their article (what does this mean?). If published, this will include your full peer review and any attached files.

Reviewer #2: **Yes: **ASAD UL ISLAM KHAN

---

## [Editor Report · Acceptance letter]

21 Nov 2024

PONE-D-24-22736R1 

PLOS ONE

Dear Dr. Elgarhy, 

I'm pleased to inform you that your manuscript has been deemed suitable for publication in PLOS ONE. Congratulations! Your manuscript is now being handed over to our production team.

Kind regards, 

on behalf of

Dr. Inés P. Mariño 

Academic Editor

PLOS ONE